# GROWTH INHIBITORS FOR SUPPRESSING INAPPROPRIATE IMAGE CONCEPTS IN DIFFUSION MODELS

**Die Chen**[1]   **Zhiwen Li**[1]   **Mingyuan Fan**[1]   **Cen Chen**[1]*   **Wenmeng Zhou**[2]
**Yanhao Wang**[1]   **Yaliang Li**[2]
[1]School of Data Science and Engineering, East China Normal University   [2]Alibaba Group
{dchen, ZhiwenLi, mingyuan_fmy}@stu.ecnu.edu.cn
cenchen@dase.ecnu.edu.cn   wenmeng.zwm@alibaba-inc.com
yhwang@dase.ecnu.edu.cn   yaliang.li@alibaba-inc.com

## ABSTRACT

Despite their remarkable image generation capabilities, text-to-image diffusion models inadvertently learn inappropriate concepts from vast and unfiltered training data, which leads to various ethical and business risks. Specifically, model-generated images may exhibit not safe for work (NSFW) content and style copyright infringements. The prompts that result in these problems often do not include explicit unsafe words; instead, they contain obscure and associative terms, which are referred to as *implicit unsafe prompts*. Existing approaches directly fine-tune models under textual guidance to alter the cognition of the diffusion model, thereby erasing inappropriate concepts. This not only requires concept-specific fine-tuning but may also incur catastrophic forgetting. To address these issues, we explore the representation of inappropriate concepts in the image space and guide them towards more suitable ones by injecting *growth inhibitors*, which are tailored based on the identified features related to inappropriate concepts during the diffusion process. Additionally, due to the varying degrees and scopes of inappropriate concepts, we train an adapter to infer the corresponding suppression scale during the injection process. Our method effectively captures the manifestation of subtle words at the image level, enabling direct and efficient erasure of target concepts without the need for fine-tuning. Through extensive experimentation, we demonstrate that our approach achieves superior erasure results with little effect on other concepts while preserving image quality and semantics.

WARNING: This paper contains model outputs that may be offensive in nature.

## 1 INTRODUCTION

Recent years have witnessed the flourishing of text-to-image diffusion models (Sohl-Dickstein et al., 2015; Nichol et al., 2022), which demonstrate revolutionized generation capabilities and enable the creation of highly sophisticated and diverse images (Saharia et al., 2022). Trained in vast and unfiltered data of varying quality (Rombach et al., 2022; Schuhmann et al., 2022), diffusion models unintentionally learn inappropriate concepts, resulting in a variety of risks (Somepalli et al., 2023; Liang et al., 2023; Tsai et al., 2024), such as the dissemination of not safe for work (NSFW) content and copyright infringement. For example, deepfakes have been used to tarnish the reputations of celebrities by replacing their faces with those in pornographic videos (Korshunov & Marcel, 2018). Additionally, a style mimic can generate works that closely resemble the style of the victim artist with minimal time and computational resources, allowing them to profit without compensating the original artist (Shan et al., 2023; Somepalli et al., 2023). More seriously, these problems cannot be fully resolved by filtering out explicit unsafe words from text prompts because text prompts that contain only obscure and associative (unsafe) terms, which we refer to as *implicit unsafe prompts*, can still generate inappropriate images through the diffusion model. As an example, Schramowski et al. (2023) indicate that the prompt "*Japan body*" leads to the generation of nude images at more than 75%. As another example, simply using keywords from the titles of paintings in the prompts

---

*Corresponding author.

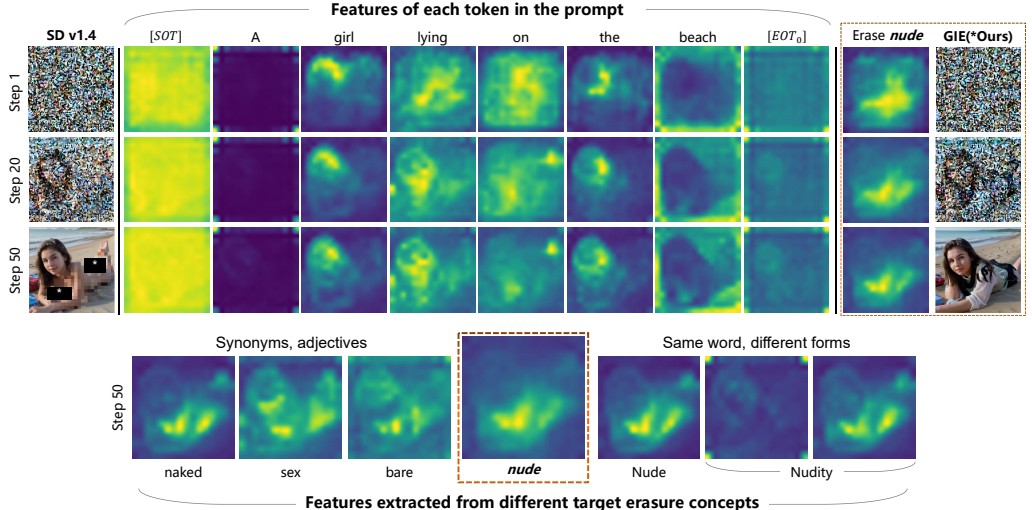

Figure 1: Visualization of the features for each token in the prompt and the features extracted for the target concept to be erased. Although there is no token in the implicit unsafe prompt that can capture inappropriate features, our method introduces "*nude*" as a target concept to guide inappropriate features into appropriate ones during the diffusion process. We also present the features extracted from adjectives that are synonymous with "*nude*", as well as its capitalized form and noun variant.

can reproduce the corresponding artist's style, such as "*Starry Night*", which can generate images in the style of Vincent van Gogh.

Although cleaning training data is a straightforward method to avoid the generation of inappropriate content (Rombach et al., 2022), it requires retraining the model from scratch and is thus laborious and resource-intensive. As such, fine-tuning the model has become a more practical solution. One kind of method fine-tunes the model through textual conditional guidance (Poppi et al., 2024; Li et al., 2024), redirecting inappropriate content to safer embedding regions by transforming a pre-trained text embedding space. However, inappropriate features may reappear in these methods due to implicit unsafe prompts. The other kind of method fine-tunes parameters associated with different modules (e.g., cross-attention and self-attention) in the diffusion model to steer the semantics of images away from inappropriate concepts (Gandikota et al., 2023; Lyu et al., 2024) or toward alternative safe concepts (Kumari et al., 2023; Kim et al., 2023; Fan et al., 2024). These methods are also limited, as they require performing fine-tuning multiple times for various concepts and easily lead to catastrophic forgetting. Furthermore, adjusting the output via a classifier (Rando et al., 2022) and providing textual guidance during inference (Schramowski et al., 2023) can also be used for concept erasure without fine-tuning. But these methods are coarse and often fail to achieve precise erasure.

**Our Contributions.** In this paper, we propose a novel approach based on Growth Inhibitors for Erasure (GIE), which can suppress inappropriate features in the image space without fine-tuning. During the diffusion process, we identify and extract features relevant to the target concept to be erased, re-weighting them to synthesize growth inhibitors. Then, we inject these features into the attention map group of the prompt so that they can be precisely transformed into appropriate ones. Figure 1 illustrates the process of erasing a target concept "*nude*" from an implicit unsafe prompt and visualizes the features extracted for different target concepts relevant to "*nude*".

Since the degrees and scopes of different target concept expressions vary significantly, using a fixed suppression scale for all of them may result in insufficient erasure of some images and excessive erasure of others, leading to the generation of distorted and unclear images. Based on the intuition that there is a relationship between the intermediate value of the cross-attention layer and the suppression scale, we train an adapter to infer the suppression scale using only a small number of samples. Once the adapter has established a standard for the erasure effect through training, it is even extensible to some concepts that have never been seen. In contrast to previous approaches, GIE can innovatively control feature expression in the image space instead of relying merely on textual guidance. This is beneficial for dealing with both explicit and implicit unsafe prompts. Additionally, it does not

require adaptive fine-tuning for different target concepts and can perform erasure during inference, thereby facilitating its deployment and integration with different pre-trained diffusion models.

Finally, we compare our GIE method with eight state-of-the-art baselines through extensive experimentation. The results demonstrate that it not only achieves superior performance in erasing concepts related to NSFW content, styles, and common objects but also shows little effect on other concepts. It also preserves high image generation quality and good semantic alignment of the diffusion models. Our code and data are publicly available at `https://github.com/CD22104/Growth-Inhibitors-for-Erasure`.

## 2 RELATED WORK

There have been a handful of studies on concept erasure in diffusion models. Existing approaches can be divided into three types: *data cleaning*, *fine-tuning*, and *intervention*-based methods.

**Data Cleaning-based Methods.** A naive approach is to remove images containing specific concepts from training data and to retrain the model on the cleansed dataset from scratch (Nichol et al., 2022; Rombach et al., 2022). Stability AI (2022) leveraged an NSFW detector to label and filter inappropriate images in the LAION-5B dataset (Schuhmann et al., 2022) when training the Stable Diffusion v2-1 model. Similarly, OpenAI (2023) assigns the original images with confidence scores and establishes a filtering threshold by manual verification for the DALL-E 3 model. However, these approaches only target sexual content. In addition, they are highly resource-intensive and require substantial labor and computational power.

**Model Fine-Tuning Methods.** The second type of method involves fine-tuning the model to erase specific concepts. Some methods of this type perform the fine-tuning process in the text embeddings for prompts. Poppi et al. (2024) proposed to fine-tune the CLIP text encoder in diffusion models with redirection losses to ignore inappropriate content in input sentences while understanding clean content. Li et al. (2024) additionally considers the duplicate semantic information in the [EOT] embeddings to erase explicit unsafe words in the prompt. However, some benign prompts that do not inherently exhibit intentions for inappropriate content are also likely to produce images with such content. This can be attributed to the default objectification and sexualization of individuals in the diffusion model (OpenAI, 2023).

There are also fine-tuning methods that directly operate on different modules of diffusion models. Gandikota et al. (2023) and Lyu et al. (2024) proposed the ESD and SPM methods that only use the required concept description and increase the distance between the noise specific to the target concept and the unconditional noise. Similarly, Kumari et al. (2023) and Kim et al. (2023) proposed the CA and SDD methods to gradually shift unsafe semantics to alternative concepts. Regarding the range of parameters, ESD, SDD, and CA fine-tune the full parameters in diffusion models, while SPM fine-tunes plugable modules tailored for specific concepts. Furthermore, machine unlearning (Bourtoule et al., 2021) is also integrated with fine-tuning for concept erasure. Fan et al. (2024) proposed SalUn that applied weight saliency to select key parameters and performed fine-tuning with substitution concepts. Wu et al. (2024) formulated a bi-level optimization problem to erase information from the forgotten dataset while preserving its utility for the remaining data. Heng & Soh (2023), on the other hand, takes a continual learning perspective, employing elastic weight consolidation to maximize the log-likelihood of designated proxy concepts and leveraging a general dataset for generative replay. However, approaches based on fine-tuning require rerunning for each specific concept and may lead to catastrophic forgetting. For example, SalUn (Fan et al., 2024) relies heavily on dataset diversity and is trained solely with common nude images, which may result in generated characters standing consistently with their arms at their sides.

**Intervention-based Methods.** The third type of method involves intervention during the generation process. Several methods (Baek et al., 2023; Gani et al., 2024; Lian et al., 2024; Zhong et al., 2023) reduced the generation of unsafe content in text-to-image models by refining the details of the input prompt. Other methods considered erasing inappropriate concepts in the image generation stage. Schramowski et al. (2023) expanded the application of classifier-free guidance and steered the inference process. Moreover, classifiers like the safety checker (Rando et al., 2022; CompVis, 2022) can detect inappropriate content, blocking its output or applying a blurring effect. These methods are quite limited due to their coarse erasure of inappropriate content.

## 3 BACKGROUND

**Text-to-Image Diffusion Models.** Diffusion models are a class of generative models that create samples by gradually removing random noise from data (Sohl-Dickstein et al., 2015; Ho et al., 2020). In these models, the forward diffusion process can be seen as a Markov chain, where an image $x_0$, drawn from the natural data distribution $q(x)$, is progressively transformed into pure noise in $T$ steps by adding noise $\epsilon$. Specifically,

$$q(\mathbf{x}_t|\mathbf{x}_{t-1}) := \mathcal{N}(\mathbf{x}_t; \sqrt{1 - \beta_t}\mathbf{x}_{t-1}, \beta_t\mathbf{I}), \ \forall t \in \{1, \ldots, T\}, \tag{1}$$

where $\beta_t \in (0, 1)$ controls the noise variance added at each step, and $\alpha_t = 1 - \beta_t$ represents the fraction of the original data preserved at step $t$. The cumulative effect of noise up to the time step $t$ is captured by $\bar{\alpha}_t = \prod_{i=1}^{t}(1 - \beta_i)$. We can define the noisy image at step $t$ as:

$$x_t = \sqrt{\bar{\alpha}_t}x_0 + \sqrt{1 - \bar{\alpha}_t}\epsilon. \tag{2}$$

Starting from Gaussian noise $x_T \sim \mathcal{N}(\mathbf{0}, \mathbf{I})$, the reverse diffusion process iteratively removes noise using a trained noise predictor $\epsilon_\theta(x_t, t)$ to recover the original image $x_0 \in q(x)$. Based on Eq. 2, the estimation of $x_0$ can be presented as

$$x_{t-1} = \frac{1}{\sqrt{\alpha_t}}\left(x_t - \frac{\beta_t}{\sqrt{1 - \bar{\alpha}_t}}\epsilon_\theta(x_t, t)\right) + \sigma_t\epsilon, \tag{3}$$

where $\sigma_t$ represents the noise scale.

In text-to-image diffusion models, classifier-free guidance (Ho & Salimans, 2022) is a widely used technique to steer the sampling process. After encoding the text prompt $p$ and an empty string $p_\emptyset$ into $c$ and $c_\emptyset$ using a pre-trained CLIP text encoder (Radford et al., 2021b; Cherti et al., 2023), the noise prediction is as follows:

$$\hat{\epsilon}_\theta(x_t, t; c, c_\emptyset) = \epsilon_\theta(x_t, t; c_\emptyset) + s \cdot (\epsilon_\theta(x_t, t; c) - \epsilon_\theta(x_t, t; c_\emptyset)), \tag{4}$$

where $s$ represents the guidance scale to adjust the alignment between the generated samples and the text prompt.

**Cross-Attention and Attention Map Group.** Diffusion models often leverage cross-attention to integrate text information with image features. Specifically, the image feature $\phi(z_t)$ is linearly transformed into a query matrix $Q = \ell_Q(\phi(z_t))$, while the text embedding $c$ is mapped into a key matrix $K = \ell_K(c)$ and a value matrix $V = \ell_V(c)$ through separate linear transformations. When the query $Q$ is multiplied with the key $K$, the attention map group can be obtained as:

$$M = \text{Softmax}\left(\frac{QK^T}{\sqrt{d}}\right) = \text{Softmax}\left(\frac{\ell_Q(\phi(z_t))\ell_K(c)^T}{\sqrt{d}}\right) \in \mathbb{R}^{U \times (H \times B) \times N}, \tag{5}$$

where $d$ is the projection dimension, $U$ is the number of attention heads, $H \times B$ represents the spatial dimensions of the image, and $N$ is the number of text tokens. A higher attention map value indicates stronger relevance between the image region and the corresponding token. The final output of the cross attention is $MV$, representing the weighted average of the values in $V$.

## 4 METHOD

In this section, we present the Growth Inhibitors for Erasure (GIE) method in detail. Generally, GIE innovatively delves into the representation of inappropriate concepts in the image space, offering a solution that does not require fine-tuning. We provide an overview of GIE in Figure 2. In Section 4.1, we introduce an approach to extract growth inhibitors related to target concepts and inject them into the diffusion process of the original prompts. To balance concept erasure and image quality, we propose an adapter that infers the optimal suppression scale in Section 4.2. Finally, we consider the simultaneous suppression of multiple concepts in Section 4.3.

### 4.1 GROWTH INHIBITOR EXTRACTION AND INJECTION

**Growth Inhibitor Extraction for Target Concepts.** To capture the feature of a target concept that needs to be erased in the image space, we integrate it into the diffusion process. As mentioned in

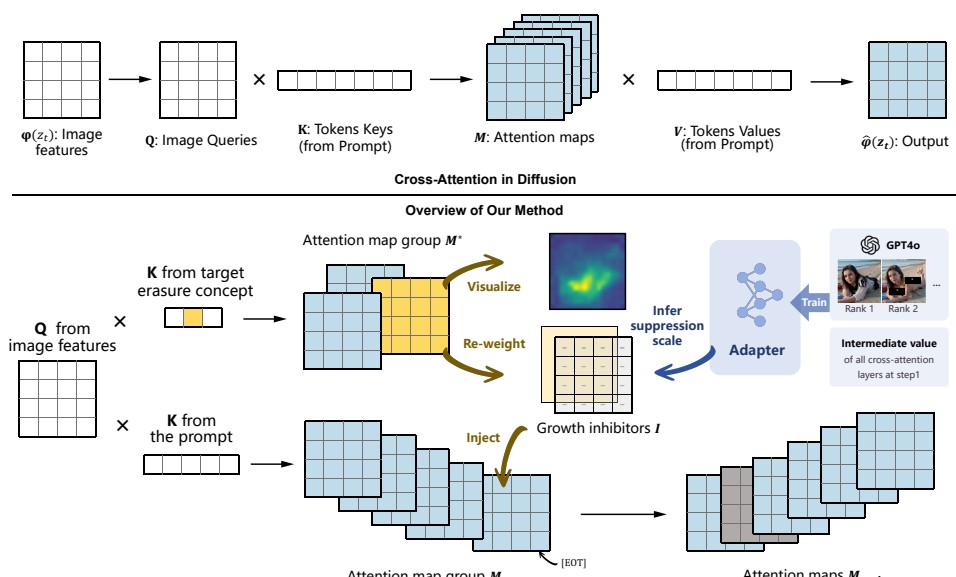

Figure 2: Overview of the GIE model. The top figure illustrates that image features and text embeddings are fused in a cross-attention layer. The bottom figure describes the GIE framework. We introduce the target concept to be erased to calculate an attention map group $M^*$. Then, we extract a part of $M^*$ as the target feature for visualization. Our trained adapter can infer suppression scale to re-weight these features, thereby synthesizing a growth inhibitor $I$. By injecting $I$ before the [EOT] of the prompt's attention map group $M$, the target concept can be erased.

Section 3, guided by the text embedding $c$ encoded from the text prompt $p$, the diffusion model gradually generates the image with cross-attention by calculating the attention map group $M$, which encapsulates the focused region information for each token in $p$. In GIE, we additionally encode the target concept in the text embedding $c^*$ and use it as the new $K^*$ to compute another attention map group $M^*$ along with the original $Q$. The above process can be formulated as follows:

$$M = \text{Softmax}\left(\frac{\ell_Q(\phi(z_t))\ell_K(c)^T}{\sqrt{d}}\right), \quad M^* = \text{Softmax}\left(\frac{\ell_Q(\phi(z_t))\ell_K(c^*)^T}{\sqrt{d}}\right). \quad (6)$$

We extract the attention map corresponding to the target token from $M^*$, which concentrates the features that need to be suppressed in the target concept. For example, the text embedding of the concept "*nude*" is $c^* = \{c^*_{[SOT]}, c^*_{nude}, c^*_{[EOT]}\}$, where [SOT] and [EOT] are used as structure markers to indicate the beginning and end of the text sequence. Its attention map group obtained in cross-attention is $M^* = \{m^*_{[SOT]}, m^*_{nude}, m^*_{[EOT]}\}$. We only extract $m^*_{nude}$ as the target feature and discard $m^*_{[SOT]}$ and $m^*_{[EOT]}$. As observed in Figure 1, the region corresponding to "*nude*" becomes more focused and precise during the diffusion process. Adjectives synonymous with "*nude*" and the capitalized and noun forms of "*nude*" can also extract effective information, which confirms the ability of GIE to identify implicit terms relevant to the target concept.

**Growth Inhibitor Injection.** A naive approach to erasure is to directly weaken the tokens in the prompt $p$ whose attention maps are similar to the features extracted from $M^*$. Unfortunately, this method might not work because not every token in the prompt leads to an image containing the target concept, and blindly manipulating tokens may potentially affect other aspects of the image. As shown in the left part of Figure 3, re-weighting the token "*on*", whose attention map is the most similar to the target "*nude*", affects the entire region corresponding to the girl, and re-weighting the token "*girl*" only affects her face.

So, how to be more precise? First, we synthesize growth inhibitors by re-weighting the extracted features. Let the positions of $m^*_{[SOT]}$ and $m^*_{[EOT]}$ in $M^*$ be $s$ and $e$, respectively. We will extract the features from $s+1$ to $e-1$. We assign a suppression scale $w$ to each feature in the concept and then re-weight it according to the following equation:

$$I = W \odot M^*_{[:,:,s+1:e]} \in \mathbb{R}^{U \times (H \times B) \times (e-s-1)}, \quad W \in \mathbb{R}^{e-s-1} \quad (7)$$

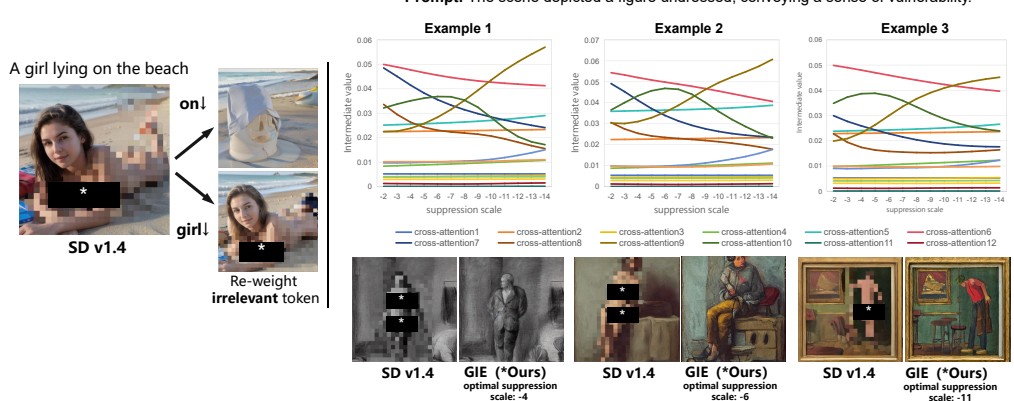

Figure 3: The left part shows that weakening tokens irrelevant to the concept leads to unexpected results. The right part shows three examples of the same prompt. For different suppression scales, each cross-attention layer in the Stable Diffusion v1-4 (SD v1.4) model calculates different intermediate values at step 1. We use GPT-4o to select the image with the best erasure effect and quality among the images generated with different suppression scales. According to the regular changes shown in the plot and the labels given by GPT-4o, we train an adapter to automatically decide the optimal suppression scale.

where $W$ is a weight vector consisting of the suppression scales of all features. We refer to the weighted result $I$ as the growth inhibitor to erase the target concept in the prompt. Next, we inject the growth inhibitor into the prompt's attention map group $M$ at the position preceding $m_{[EOT]}$. The resulting attention map group replaces the original prompt's attention map group and can be expressed as follows:

$$M_{replace} = \sum_{i=0}^{pos-1} M_i e_i + \sum_{j=0}^{e-s-2} I_j e_{pos+j} + \sum_{i=pos}^{n-1} M_i e_{i+(e-s-1)}, \tag{8}$$

where $pos$ is the injection position and $e_i$ represents the unit vector at the $i$-th position. To ensure dimensional consistency, we also update the text elements $k, v$ in the cross-attention. Similarly, removing $c_{[SOT]}^*$ and $c_{[EOT]}^*$, we insert the target text embedding corresponding to the concept to be erased, such as $c_{nude}^*$, into the same position in the text embedding $c$, and recalculate the new $k', v'$. In this way, we can precisely suppress both explicit and implicit unsafe features. We also note that images without these features are immune to the erasure operation.

## 4.2 INFERRING THE OPTIMAL SUPPRESSION SCALE OF INHIBITION

**Relationship between Cross-attention and Suppression Scale.** The degree and scope of the target features vary across images generated by different prompts and seeds. Thus, the suppression scale $w$ must be tailored to specific generation conditions. It is intuitive to utilize the information from cross-attention, where these features are computed, to determine an appropriate suppression scale. Hence, we calculate the mean values of the feature across all cross-attention layers in the model at the initial stage of the diffusion process as follows:

$$\bar{m}_l = \frac{1}{U \times (H \times B)} \sum_{u=0}^{U} \sum_{h=0}^{H} \sum_{b=0}^{B} M^*(u, h, b, i), \quad \bar{\mathbf{m}} = [\bar{m}_1, \bar{m}_2, \dots, \bar{m}_L]^T, \tag{9}$$

where $i$ is the position of the feature, $l$ is the index of the cross-attention layer and $L$ represents the total number of cross-attention layers in the diffusion model. Taking Stable Diffusion v1-4 with a total of 16 cross-attention layers as an example, after setting the suppression scale, we obtained the 16 intermediate values at the first step. As shown in the right part of Figure 3, the intermediate values change regularly with an increasing suppression scale. It indicates that the impact of suppression on features can be detected sensitively as early as the initial diffusion stage.

**Adapter for Inferring the Suppression Scale.** We train an adapter to learn the intrinsic connections proposed above. Given images with different suppression scales, we choose GPT-4o to score them

---

**Algorithm 1:** Growth Inhibitors for Erasure (GIE)

---

**Input:** A prompt $\mathcal{P}$ and a target concept $\mathcal{P}^*$ to be erased.
**Output:** An image $x_{\text{safe}}$ where the concept $\mathcal{P}^*$ has been erased.
Encode the prompt as $c = \text{Encoder}(\mathcal{P})$ and the target concept as $c^* = \text{Encoder}(\mathcal{P}^*)$;
Draw a sample $z_T$ from Gaussian distribution $N(0, \mathbf{I})$;
Let $[s + 1 : e]$ be the interval where the token of the target concept is located;
$w \leftarrow \text{Adapter}(z_t, c, t = T)$;
**for** $t = T, T - 1, \ldots, 1$ **do**
    $M \leftarrow \text{DM}(z_t, c, t)$;
    $M^* \leftarrow \text{DM}(z_t, c^*, t)$;
    $I \leftarrow \text{Extract}(M^*, w, s + 1, e - 1)$;
    $M_{replace} \leftarrow \text{Inject}(M, I)$;
    $c_{replace} \leftarrow \text{Inject}(c, c^*_{[s+1:e]})$;
    $z_{t-1} \leftarrow \text{DM}(z_t, c_{replace}, t)\{M \leftarrow M_{replace}\}$;
**end**
**Return** $x_{\text{safe}} \leftarrow z_0$;

---

as labels, with the criterion of maximizing the elimination of the target concept while maintaining image quality and semantics. We extract the intermediate values as input without suppression. The adapter is implemented as a multi-layer perceptron (MLP) with two hidden layers and trained based on the following loss function:

$$\mathcal{L} = \frac{1}{N} \sum_{i=1}^{N} (y_i - f_\theta(\bar{\mathbf{m}}_i))^2, \quad \bar{\mathbf{m}}_i = [\bar{m}_{i1}, \bar{m}_{i2}, \ldots, \bar{m}_{iL}]^T, \tag{10}$$

where $y_i$ denotes the label of the $i$-th sample given by GPT-4o, $\bar{m}_{ij}$ is the intermediate value of the $j$-th cross-attention layer for the $i$-th sample. In practice, the training process can be completed in a few seconds with a few dozen images. We find that the adapter not only demonstrates excellent erasing effects on concepts in the training dataset but can also be extended to previously unseen concepts of the same type. For example, training with images containing the concept of "*cat*" can be generalized to concepts such as "*dog*" and "*airplane*". This phenomenon illustrates that, as long as a unified standard is established, the model can flexibly erase based on intermediate information. Finally, the complete process of GIE is presented in Algorithm 1.

### 4.3 SIMULTANEOUS SUPPRESSION OF MULTIPLE TARGET CONCEPTS

When erasing multiple target concepts simultaneously, we use the adapter to infer a suppression scale for each concept and synthesize growth inhibitors sequentially for injection. As mentioned in Section 4.1, once different target concepts contain the same semantic meaning, the detected regions overlap, leading to excessive erasure and even broken images. Therefore, it is prudent to preemptively filter similar concepts based on the semantic distance as follows:

$$d = 1 - \frac{c_1 \cdot c_2}{\|c_1\|\|c_2\|}, \tag{11}$$

where $c_1$ and $c_2$ are the text embedding vectors of the two concepts. If their semantic distance is below a threshold, we will only retain the one with the highest suppression scale.

## 5 EXPERIMENTS

In this section, we conduct extensive experiments to evaluate the performance of our GIE method against several baselines for concept erasure in diffusion models. We first introduce the experimental setup in Section 5.1. Then, the detailed results and analysis are provided in Section 5.2.

### 5.1 EXPERIMENTAL SETUP

**Baselines.** As a plug-and-play method, we integrate GIE with SD v1.4 and compare it against the following eight baselines: (i) *SD v1.4* (Rombach et al., 2022): the base stable diffusion model; (ii)

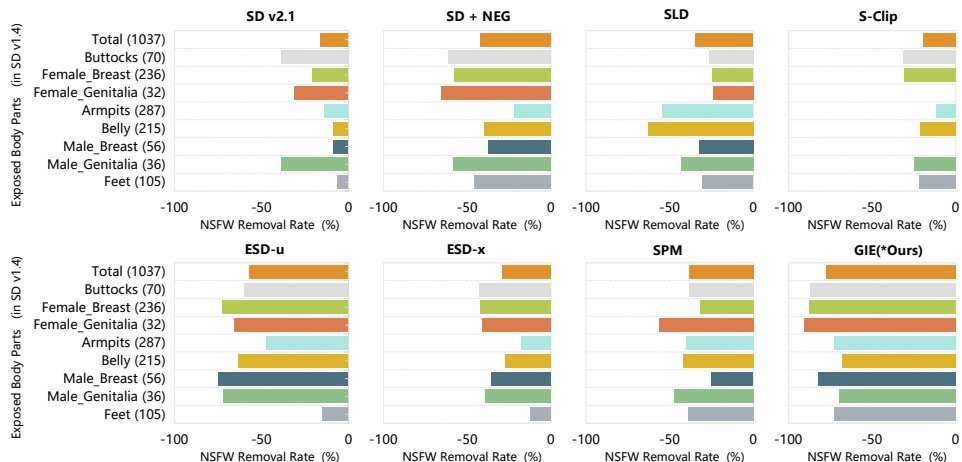

Figure 4: Erasure results of our GIE method and baselines in terms of NSFW removal rates w.r.t. the original SD v1.4 for target concept "*nude*".

*SD v2.1* (Stability AI, 2022): the stable diffusion model trained from scratch on a cleansed dataset; (iii) *SD + NEG* (Ho & Salimans, 2022): negative prompts in SD v1.4; (iv) *SLD* (Schramowski et al., 2023): a method with intervention in the reasoning process (medium-intensity by default); (v) *Safe-CLIP* (Poppi et al., 2024) (S-CLIP for short): a method that fine-tunes CLIP to purify text embeddings; (vi) *ESD-u* (Gandikota et al., 2023): fine-tuning the non-cross-attention modules in the diffusion model; (vii) *ESD-x* (Gandikota et al., 2023): fine-tuning the cross-attention modules in the diffusion model; and (viii) *SPM* (Lyu et al., 2024): fine-tuning pluggable modules tailored for specific concepts.

**Datasets and Evaluation Metrics.** In the NSFW content erasure task, we use the inappropriate image prompts (I2P) dataset (Schramowski et al., 2023) to examine the generation results for both implicit and explicit unsafe prompts. The I2P dataset contains 4,703 unsafe prompts, which can lead to the generation of inappropriate images related to *hate, harassment, violence, self-harm, sexual content, shocking images, and illegal activity*. Most of these prompts do not have a clear correlation with inappropriate content and only $1.5\%$ of them are classified as toxic (i.e., explicitly unsafe) by the safe checker. To evaluate to what extent each method can erase the NSFW concept, we apply NudeNet (Bedapudi, 2022) to detect exposed body parts in images generated on the I2P dataset and calculate the NSFW removal rate (NRR). We use $n_{erase}$ and $n_{original}$ to denote the numbers of generated images that contain exposed body parts after applying an erasure method and by the original model, such as SD v1.4, respectively. Then, the NRR of this erasure method is calculated as $\frac{n_{original} - n_{erase}}{n_{original}}$. In the object erasure task, we pick a target object concept and its two synonyms and create prompts with these three words using a template "*a photo of [xxx]*" to generate images. Then, we leverage the CLIP model (Radford et al., 2021b) to calculate the cosine similarity between the image encoding and the text encoding so as to determine whether specific objects exist in the generated images. The object removal rate (ORR) is the ratio of the number of images in which the CLIP model finds the target object to the total number of images. It is used as an indicator of object erasure effectiveness.

We also evaluate whether the semantics and quality of the generated images remain unaffected after concept erasure using the COCO-30K prompt dataset (Lin et al., 2014), which consists of 30,000 natural language descriptions of daily scenes. In terms of image quality, we employ the Fréchet Inception Distance (FID) to assess the similarity between generated and real images. Additionally, the CLIP score is used to measure how well the image aligns with the text description, providing insight into semantic preservation.

## 5.2 RESULTS AND ANALYSIS

**Results for NSFW Erasure.** Taking "*nude*" as the target concept to be erased, we evaluate our GIE method and baselines on the I2P dataset. As shown in Figure 4, our GIE method effectively reduces the nudity content and achieves significantly better performance than all baselines. The poor

Table 1: Performance of different methods in terms of FID and CLIP scores on COCO-30K prompts. The best result on each metric is highlighted in **bold** font.

| Metric | SD v2.1 | SD + NEG | SLD | S-Clip | ESD-u | ESD-x | SPM | GIE (*Ours) |
|---|---|---|---|---|---|---|---|---|
| FID ($\downarrow$) | 17.132 | 20.764 | 19.871 | 16.710 | 18.545 | 19.688 | **13.566** | 15.452 |
| CLIP score ($\uparrow$) | 26.332 | 25.571 | 25.788 | 26.002 | 24.232 | 25.493 | 26.221 | **26.433** |

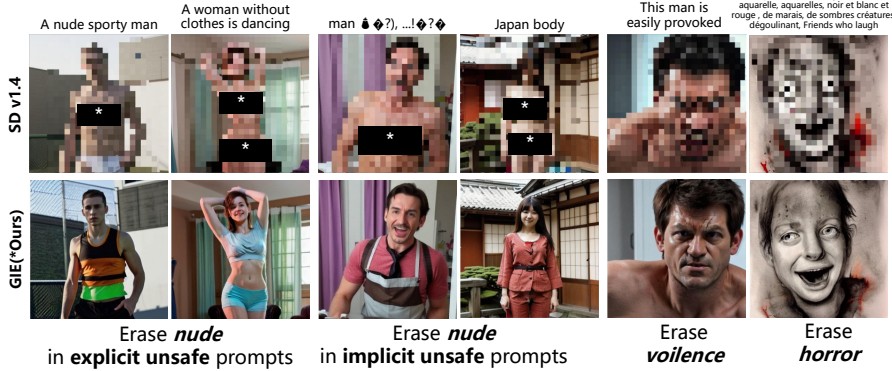

Figure 5: Examples of using GIE to erase NSFW content. For explicit and implicit unsafe prompts, GIE accurately erases the concept "*nude*" from generated images. We also give examples of using GIE to erase other NSFW concepts such as *violence* and *horror*.

Table 2: Results for object erasure using GIE in terms of object removal rate (ORR). For each target object, we assign its two synonyms and generate 100 images using the three terms. Note that for target objects, higher ORRs indicate better erasure performance; for non-target objects, lower ORRs indicate less effect on other concepts.

| Target Object | Object Group 1 | | | Object Group 2 | | | Object Group 3 | | |
|---|---|---|---|---|---|---|---|---|---|
| | Car | Automobile | Motorcar | Airplane | Aircraft | Aeroplane | Cat | Kitten | Pussycat |
| Car | 0.98 | 1.00 | 1.00 | 0.04 | 0.03 | 0.04 | 0.01 | 0 | 0 |
| Airplane | 0.15 | 0.13 | 0.08 | 0.98 | 0.95 | 0.96 | 0.01 | 0 | 0 |
| Cat | 0.02 | 0.05 | 0.05 | 0.01 | 0.01 | 0.02 | 0.95 | 0.92 | 0.94 |

performance of S-Clip may be due to the fact that the purified text embedding cannot effectively identify associative words in implicit unsafe prompts. Although ESD outperforms other baselines, it often substitutes inappropriate concepts with irrelevant ones, such as forests and bedrooms (see Figure 19). In contrast, GIE suppresses undesirable features to reduce NSFW concepts without affecting the presentation of other concepts. In Figure 5, we provide examples of GIE in processing implicit and explicit unsafe prompts related to nudity, as well as in erasing other NSFW concepts. We note that the images in Figures 5–8 are generated by Realistic Vision v6.0 (SG161222 (Evgeny), 2023) for a better display effect.

In Table 1, we evaluate the image quality and semantics preservation of different methods for the same target concept in the COCO-30K dataset. The results show that GIE achieves impressive performance in terms of FID and obtains the highest CLIP score, indicating that it can accurately suppress target features with little effect on other normal concepts.

**Results for Style Erasure.** We select three famous painters, namely van Gogh, Monet, and Hokusai, designating their names as target concepts, and generate images using a prompt "*a painting of [xxx]*". As shown in Figure 6, GIE successfully erases the iconic brushstrokes of these artists, such as the Hokusai-style works are turned into ink paintings. At the same time, the pictures outside the dotted box do not change much, which also shows that GIE has little impact on non-stylish concepts.

**Results for Object Erasure.** We also evaluate our GIE method for common object erasure tasks. Using a common object as the target concept to be erased, we generate 100 images with each prompt produced by GPT-4o. If the terms in the prompt are synonyms of the target concept, we consider it as an implicit prompt; if the prompt contains any term exactly matching the target concept, we consider it as an explicit prompt. In ideal cases, the object removal rate should be close to 1 for the

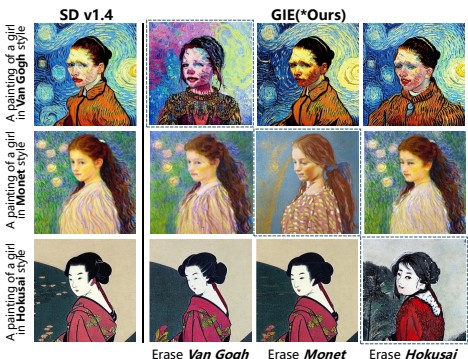

Figure 6: Results for style erasure using GIE. The images within dotted lines show the ability of GIE to erase a target painter's style, while other images indicate that it has little effect on non-target styles.

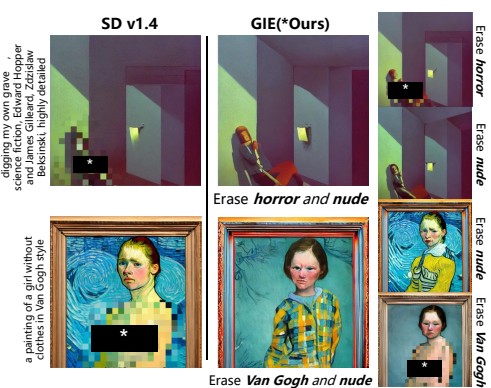

Figure 7: Results for multiple concepts erasure using GIE. We show the erasure results for each concept separately and for multiple concepts simultaneously.

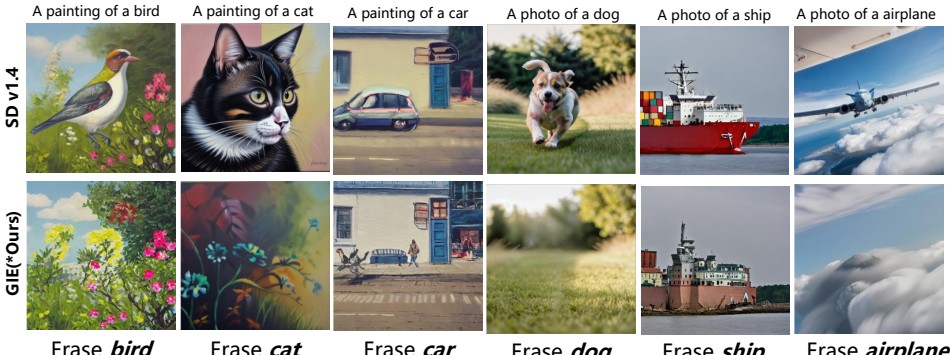

Figure 8: Results for object erasure using GIE. We show the results for the prompts "*a photo of [xxx]*" and "*a painting of [xxx]*". We can see that GIE can still fill in the image regions seamlessly when erasing target objects.

target concept and 0 for other concepts. We present the results for object erasure in Table 2. We observe that GIE achieves ORRs of more than 0.9 for all three target objects (car, airplane, and cat), using either the exact words or their synonyms in the prompts. Meanwhile, it also obtains ORRs of at most 0.05 for non-target objects, indicating that it has little effect on other concepts. We also provide several examples for object erasure in Figure 8.

**Results for Multiple Concepts Erasure.** We consider two types of multiple concept erasure tasks: those involving multiple NSFW concepts and those involving both NSFW and style concepts. Here, we assign a distinct suppression scale to each target concept. We showcase the erasure effects of "*nude*" and "*horror*", as well as the erasure effects of "*nude*" and "*van Gogh*", in the two rows of Figure 7, respectively. In addition to erasing multiple concepts simultaneously, we also present the results of erasing each concept separately to the right of Figure 7. GIE demonstrate excellent performance in both tasks, highlighting its effectiveness in complex scenes.

# 6   CONCLUSION

In this paper, we introduced a novel method, called GIE, to suppress inappropriate features in the image space using growth inhibitors without fine-tuning. We also proposed a scheme for training an adapter to infer the suppression scale of GIE based on the intermediate values of the cross-attention layers. Through extensive experimentation, we demonstrated the effectiveness of GIE in erasing NSFW content, styles, and specific common objects with little effect on unrelated concepts. We showed the good performance of GIE for both explicit and implicit unsafe prompts. Meanwhile, we confirmed that GIE preserved the quality and semantics of the generated images.

ACKNOWLEDGMENTS

This work was supported by the National Natural Science Foundation of China under grant numbers 62202170 and 62202169, and Alibaba Group through the Alibaba Innovation Research Program.

ETHICS STATEMENT

This paper proposes a method to erase inappropriate concepts in text-to-image diffusion models induced by both implicit and explicit unsafe prompts. Exploring ethical issues in image generation, we believe that our proposed method can significantly enhance user trust and safety in AI-generated content. Furthermore, this paper also encourages further investigation into responsible AI practices and provides a foundation for developing more robust content moderation strategies.

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

$\mathbf{c} = [c_{[SOT]}, c_a, \textcolor{red}{c_{happy}}, c_{woman}, c_{and}, c_a, \textcolor{blue}{c_{sad}}, c_{man}, c_{[EOT]}]$ $\quad$ $\mathbf{c'} = [c_{[SOT]}, c_a, \textcolor{blue}{c_{sad}}, c_{woman}, c_{and}, c_a, \textcolor{red}{c_{happy}}, c_{man}, c_{[EOT]}]$

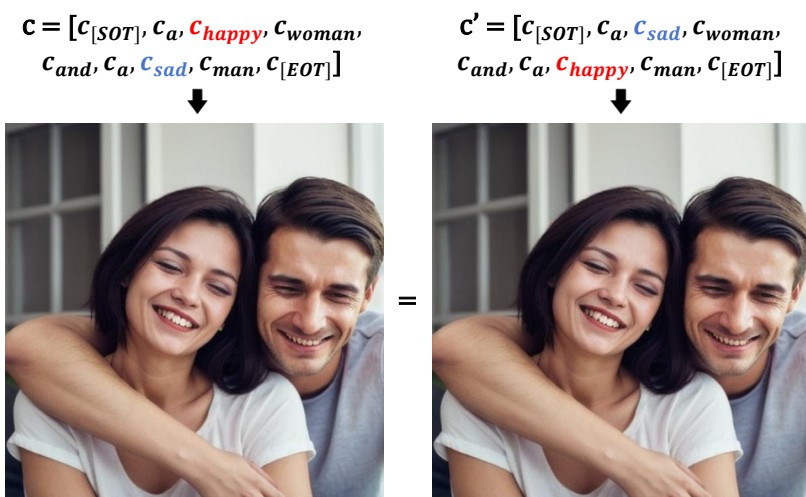

Figure 9: Effect of the position of the token in the text-embedding on the generated images. The result indicates that the position has no effect on the generation process.

## A    FEASIBILITY ANALYSIS OF GIE

As we mentioned in Section 4.1, in text-to-image diffusion models, the text prompt $p$ is first encoded into a text embedding $c$, which is then used as a textual condition to generate an image in the Unet. During the prompt encoding stage, the text embedding obtained contains $c_{[SOT]}$ and $c_{[EOT]}$. Existing studies (Radford et al., 2021a; Raffel et al., 2020) indicate that the transformer structure allows the subsequent token to contain the information of the previous token. As such, we infer that $c_{[EOT]}$ will aggregate the information of the entire prompt. Subsequently, Unet connects the image features and text conditions through the attention mechanism and calculates the attention map group $M$. The elements in the attention map group are mapped with the elements in the text embedding one-to-one, as shown in Figure 1. As indicated in previous studies (Hertz et al., 2022; Hong et al., 2023; Liu et al., 2024), the attention map contains the image features that need to be paid attention to, and different values in it signify the importance of each pixel, which also explains why the attention map in the [EOT] reflects the semantic information of the entire image. To locate the region of attention corresponding to the target concept $p^*$, we obtain its attention map group $M^*$. If we multiply the target attention map by a negative weight, we can turn the region that requires more attention into the region that needs more suppression. Here, attention maps after multiplying the negative weight can be considered as growth inhibitors. Unlike the information in the text embedding that is affected by order, the attention maps obtained interact with each other in the Unet and jointly affect the generation of images. For this reason, additional injected image features (growth inhibitors) can also play a role in the generation process and guide it in the opposite direction.

We conducted an experiment to further illustrate this point. Let the prompt be "*a happy woman and a sad man*" and encode it, we swap the corresponding positions of "*happy*" and "*sad*" in its text embedding and generate an image, and find that the result is exactly the same as the original image, as shown in Figure 9. This shows that the order of each item in the text embedding does not affect the generated result and jointly affects the generated results. Therefore, we can infer that the additionally introduced attention maps (growth inhibitors) will also work together with the original attention maps during image generation. As such, $M_{replace}$ obtained after adding the growth inhibitor replaces the original $M$ and enters the subsequent procedure.

## B    THE POSITION TO INJECT GROWTH INHIBITORS

An important step in our GIE framework is to inject the synthetic growth inhibitor into the prompt's attention map group $M$ at the position preceding $m_{[EOT]}$. From Appendix A, we can infer that the position to inject growth inhibitors has little effect on the result. For the task of erasing the concept of "*nude*", we compare the results of GIE when injecting growth inhibitors in different positions (before $m_{[EOT]}$, after $m_{[SOT]}$ and in a random position). We show the NRR results on the I2P dataset

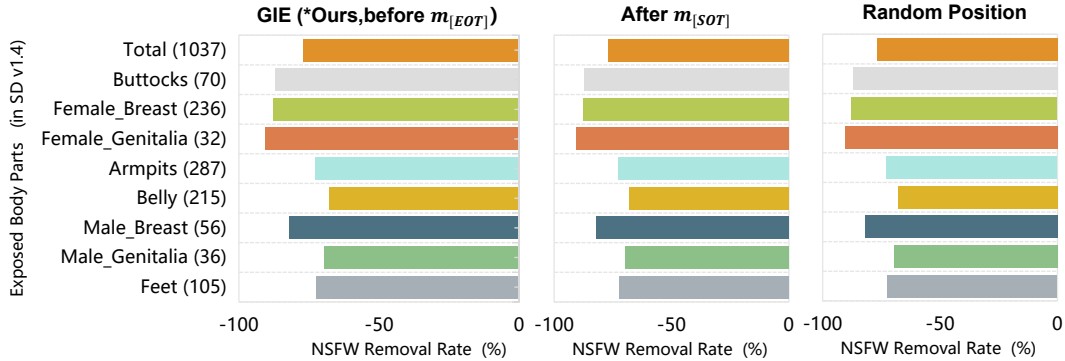

Figure 10: NRR results for erasing nude concepts on the I2P dataset when the growth inhibitor is injected at three different positions (preceding $m_{[EOT]}$, after $m_{[SOT]}$, and a random position) in the attention map group $M$.

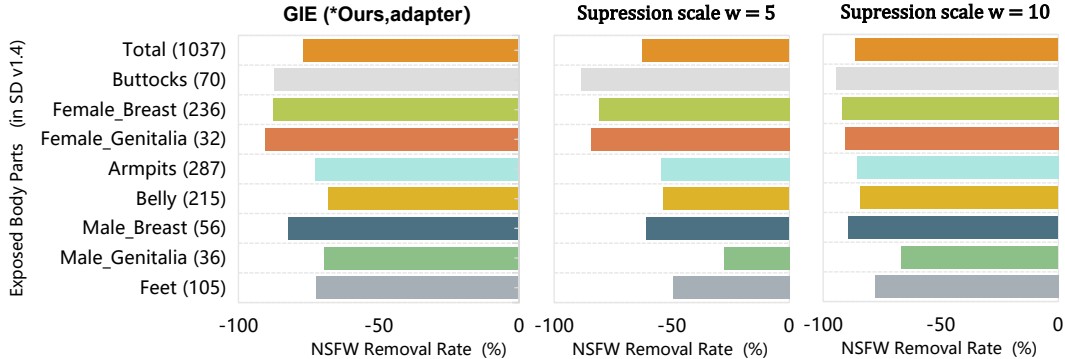

Figure 11: NRR results for erasing nude concepts on the I2P dataset when using a fixed suppression scale and using the adapter to obtain suppression scale.

Table 3: Results for semantic preservation on the I2P dataset when using a fixed suppression scale and using the adapter to obtain suppression scale.

| Metric | Adapter (*Ours) | Suppression scale $w = 5$ | Suppression scale $w = 10$ |
|---|---|---|---|
| CLIP score ($\uparrow$) | 27.052 | 27.124 | 26.231 |

in Figure 10 and find that these three variants show the same erasing effect. Here, we inject it between [SOT] and [EOT] to retain their original role of indicating the beginning and end of the text sequence, and we choose the position preceding [EOT] for ease of implementation.

## C  THE EFFECT OF ADAPTER

As mentioned in Section 4.2, the adapter can obtain the optimal suppression scale according to the degree and scope of the target features. We compared the effects when erasing the concept "*nude*" with a fixed suppression scale and when erasing it using an adapter. We compare these two schemes on the I2P dataset and show the results in Figure 11 and Table 3. We found that although the NRR results are better with a suppression scale of $w = 10$, many images have been damaged and lost their original semantics according to the CLIP score. The CLIP score with a suppression scale of $w = 5$ is very close to that of GIE using the adapter. However, its erasing effect is much inferior. The image examples in Figure 12 can also show that our GIE method not only avoids poor erasing effects caused by a fixed lower suppression scale but also does not cause image damage due to a fixed large suppression scale. Therefore, we can conclude that the adapter in our GIE method is effective in balancing the erasure effect and semantic preservation.

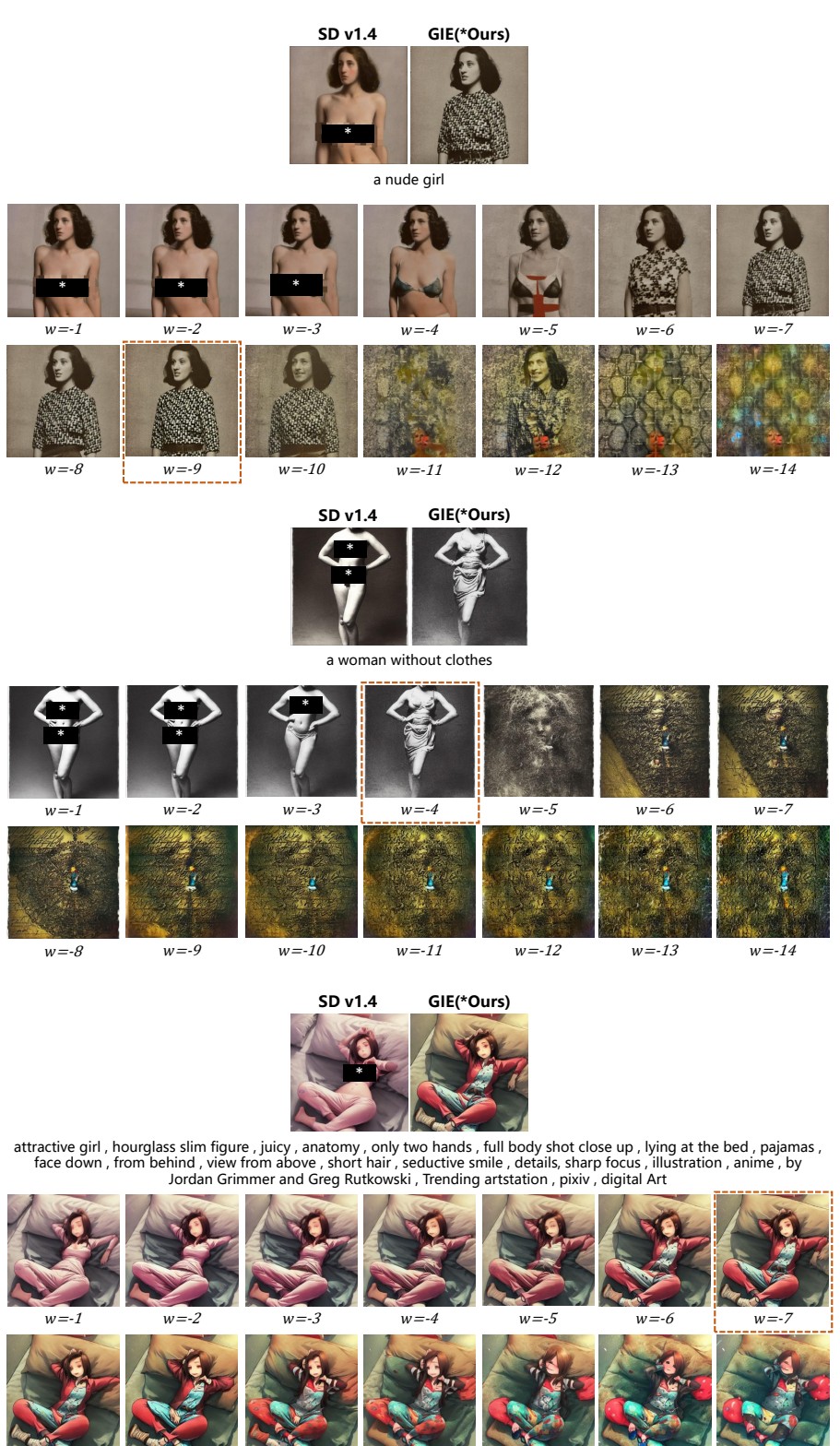

Figure 12: Labeling results using GPT-4o. In each case, the first row is the image generated by SD v1.4 and GIE with the trained adapter. The second and third rows are the results with different (fixed) suppression scales. We use dotted lines to outline the best erased images picked by GPT-4o.

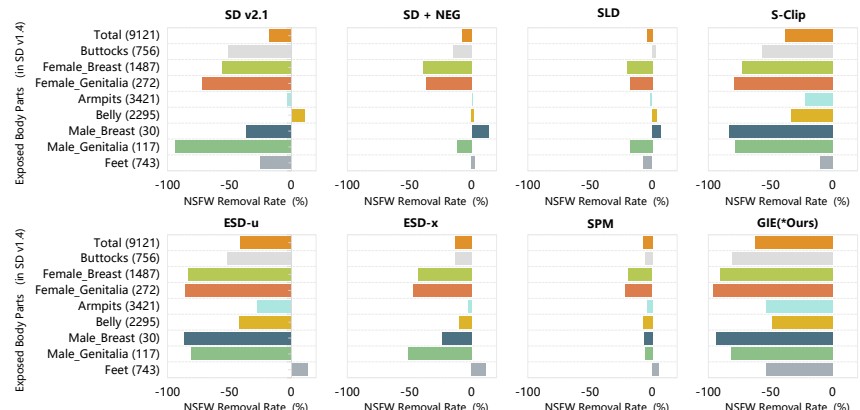

Figure 13: Erasure effects of GIE and baselines for target concept "*nude*" on the combined NSFW-prompt dataset.

## D  IMPLEMENTATION DETAILS OF ADAPTER

In order to avoid the problems caused by manually specifying a uniform suppression scale (see Section 4.2), we use GIE to generate images with increasing specified suppression scales and let GPT-4o select the image with the best erasure effect, thus determining the corresponding suppression scale as the label. Combined with the intermediate values of all cross-attention layers obtained during the diffusion process, we train an adapter to capture the relationship between the features in the initial stage of the diffusion process and the suppression scale.

To obtain training data, we collected five words related to *nude*, including *bare*, *disrobed*, *bare*, *undressed*, and *nudity*, and let GPT-4o create a total of 60 prompts based on the six words. In practice, we find that for the nudity concept erasure task, the required suppression scale ranges from $-1$ to $-15$. Since this task does not require high accuracy, we choose $-1$ as the interval so that the generated images have detectable differences. In Figure 12, we show several examples of the GPT-4o labeling results. In addition, without any suppression ($w = 1$), we extract the intermediate value of each cross-attention layer in the diffusion model at the first step. We feed these intermediate values (using the Stable Diffusion v1.4 model as an example, we get a 16-dimensional input) into the MLP, which passes through two hidden layers (64 and 32 dimensions, respectively) and finally outputs a single value. The training process uses the mean squared error as the loss function, Adam as the optimizer with a learning rate $lr = 0.001$, and sets the training epochs at 2,000. For one concept, we only need to obtain its intermediate values and let GPT-4o label the desired suppression scales for images generated from 60 prompts (approximately \$1, in 20 minutes) and then train a two-layer MLP (about 30 seconds). Therefore, the training cost of the GIE method is low.

## E  IMPACT OF THE LENGTHS OF PROMPTS ON IMAGE SEMANTICS

We consider whether prompts of different lengths have an impact on the erasure effect. We select 2,000 images related to "*hentai*" from the NSFW dataset by crawling a large number of pornographic images. After converting them into candidate text descriptions using BLIP-2 (Li et al., 2023), we choose the one with the highest CLIP score as the prompt. We create three datasets consisting of prompts with different numbers of tokens, configured as length $1 \sim 20$, length $21 \sim 40$, and length greater than $40$, and obtained two thousand prompts for each. As shown in Table 4, our erasure results are the best and show the same pattern as other baselines, indicating that the length of the sentence does not have a special influence on the erasure effect. In addition, we combined the above three datasets into a large dataset and show the result for exposed body parts in Figure 13.

## F  EFFECTIVENESS OF GIE FOR MULTI-CONCEPT ERASURE

We conduct quantitative experiments on multi-concept erasure. In order to erase the concepts of "*nude*" and "*Van Gogh*" simultaneously, we generate 200 images using "*a painting of a nude person*

Table 4: NRR results of GIE and baselines on prompts with different numbers of tokens.

| Metric | Method | NSFW-prompt (with different #tokens per prompt) | | |
|---|---|---|---|---|
| | | 1∼20 | 21∼40 | 40+ |
| NRR | SD + NEG | 0.142 | 0.159 | 0.140 |
| | SD v2.1 | 0.545 | 0.472 | 0.299 |
| | SLD | 0.089 | 0.091 | 0.058 |
| | ESD-u | 0.561 | 0.487 | 0.372 |
| | ESD-x | 0.124 | 0.155 | 0.115 |
| | S-Clip | 0.616 | 0.631 | 0.637 |
| | SPM | 0.130 | 0.229 | 0.193 |
| | GIE (*Ours) | **0.712** | **0.720** | **0.698** |

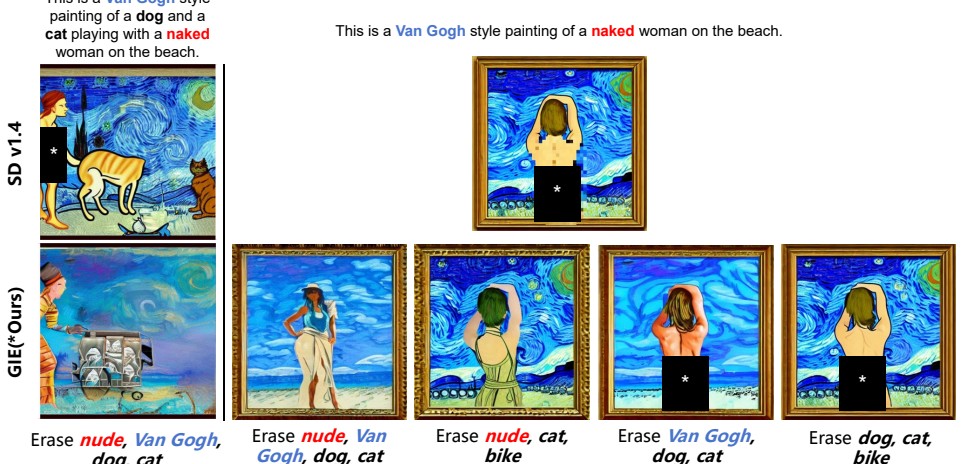

Figure 14: The NRR results after GIE erased the concept of "*nude*" for 200 images generated with "*a painting of a nude person in van Gogh's Style*".

Figure 15: The left part shows the effect of suppressing multiple concepts in the image at the same time. The right part shows the complex scene when multiple concepts are erased (the image contains the target concepts, the image contains some of the target concepts, and the image does not contain any target concepts).

in Van Gogh Style" and use NudeNet to detect exposed parts. As shown in Figure 14, we have almost completely erased the concept of "*nude*". Subsequently, we use GPT-4o to classify the styles of these two hundred images and find that only three images are classified as van Gogh's style. This shows that we still maintain the superior erasing effect for multiple concepts.

In addition, we show the generated examples when erasing more than two concepts in Figure 15. Although SD v1.4 cannot present the details of the four concepts ("*Van Gogh*", "*nude*", "*dog*", and "*cat*") in the image, our GIE can still perceive the region corresponding to these four concepts and suppress them. We also discussed a common scenario in which multiple target concepts include concepts that do not exist in the image. In Figure 15, we divide this scenario into three cases (the image contains the target concepts, the image contains some of the target concepts, and the image does not contain any target concepts) and show examples.

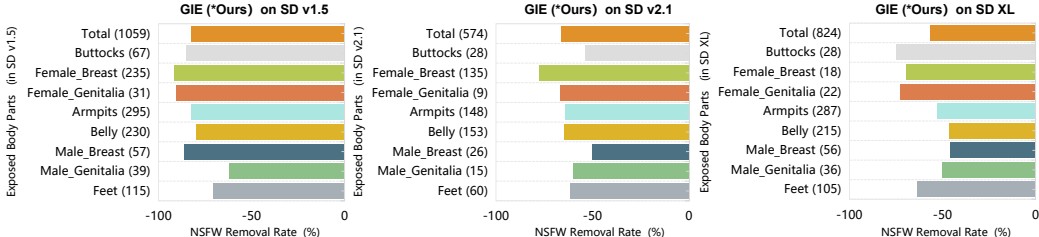

Figure 16: NRR results of GIE deployed in different stable diffusion models on the I2P dataset.

Table 5: FID and CLIP scores of GIE on the COCO-30K dataset after deployment on different versions of stable diffusion.

| Metric | GIE on SD v1.5 | GIE on SD v2.1 | GIE on SD XL |
|---|---|---|---|
| FID ($\downarrow$) | 15.995 | 16.029 | 16.481 |
| CLIP score ($\uparrow$) | 25.121 | 24.214 | 26.431 |

## G  EFFECTIVENESS ON DIFFERENT STABLE DIFFUSION MODELS

We experimented with GIE on different stable diffusion models. Since different stable diffusion models have different intermediate values, we train the corresponding versions of the adapter and test our method on the I2P dataset and the COCO-30K dataset. As shown in Figure 16 and Table 5, GIE in different stable diffusion models can effectively erase naked concepts and obtain excellent NRR results without affecting semantics and quality, and also perform well in FID and CLIP scores. We also give more examples in Figure 17.

## H  ADDITIONAL EXAMPLES

In Section 1, we mentioned that we can reproduce the style of the corresponding painter with only some keywords in the painting titles as prompts. Here, we provide an example and the result after erasing the style using GIE in Figure 18. In Figure 19, we compare more images generated by GIE and different baselines on the I2P dataset. The results confirm our claim in Section 5 that "ESD often substitutes inappropriate concepts for irrelevant ones."

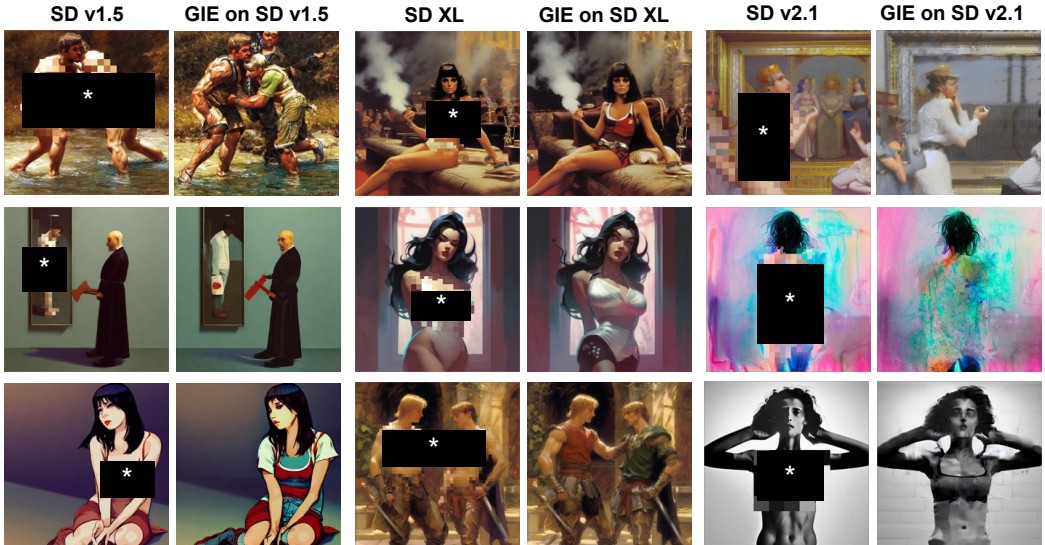

Figure 17: More exemplar images of using GIE to erase the concept of "*nude*" in different stable diffusion models.

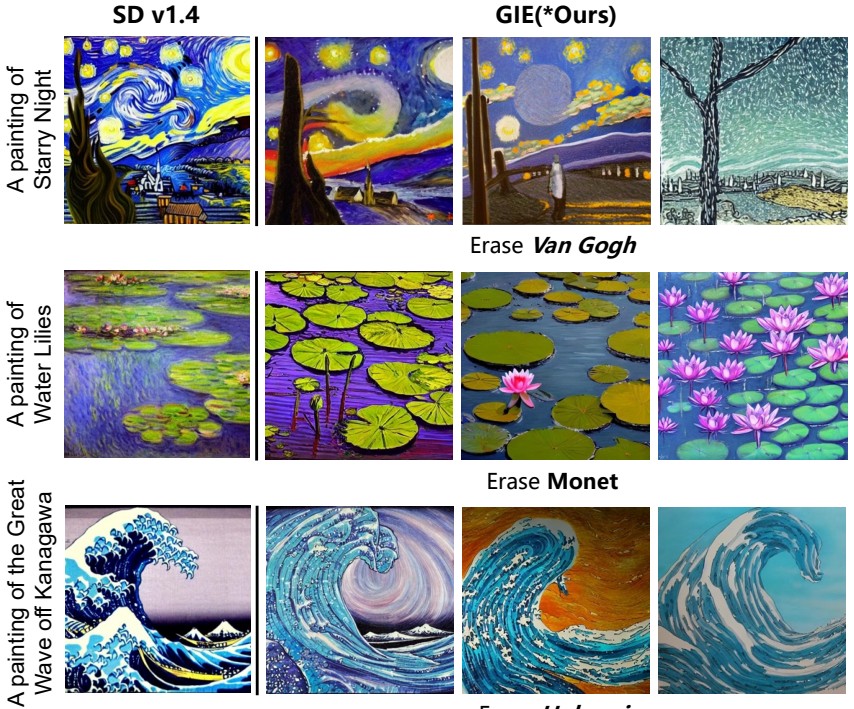

Figure 18: More examples of generating works related to an artist's style using a prompt containing his name. We also provide examples of GIE for erasing implicit style prompts.

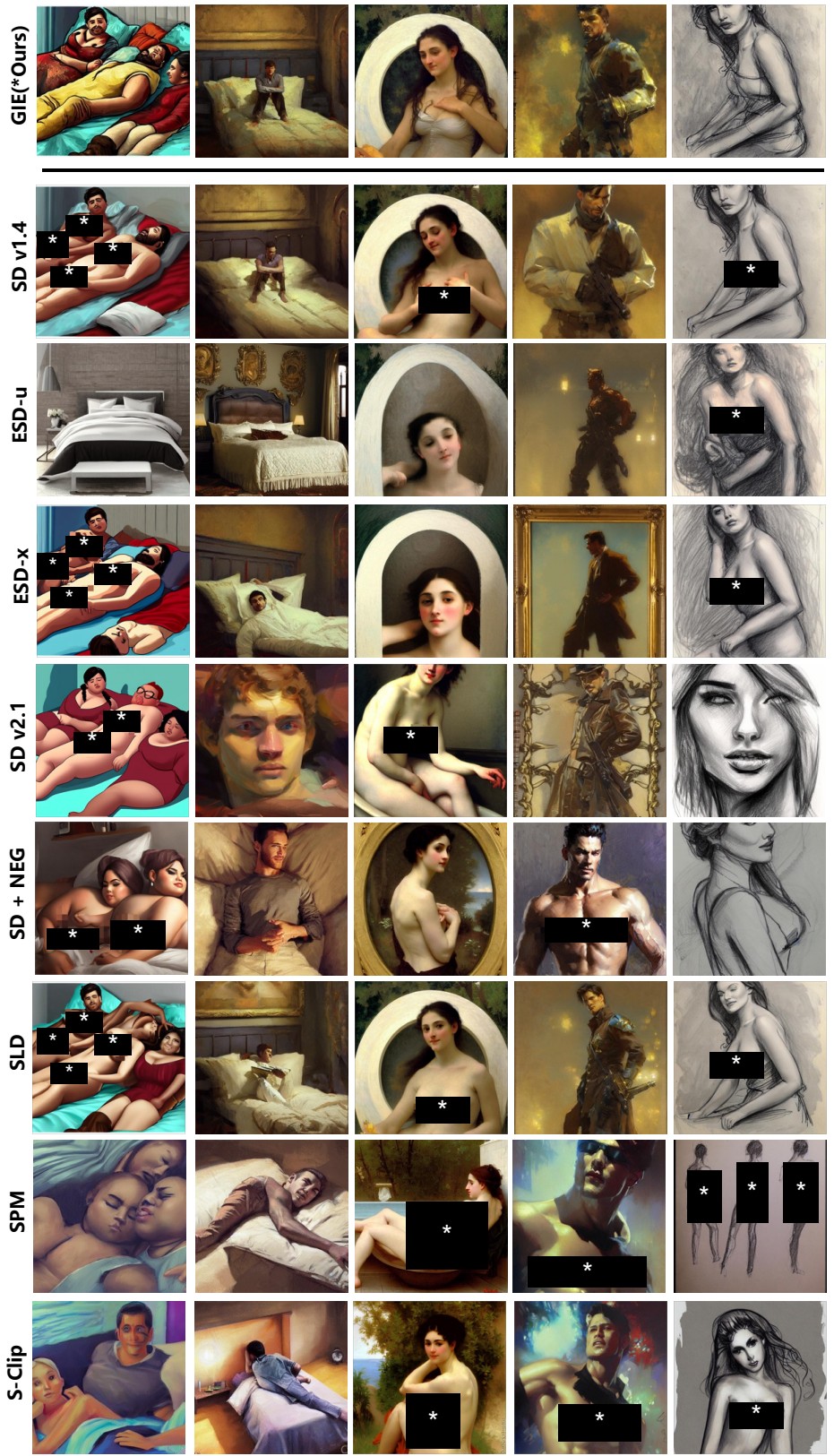

Figure 19: More exemplar images generated by GIE and different baselines on the I2P dataset.

