# OpenReview forum: "Growth Inhibitors for Suppressing Inappropriate Image Concepts in Diffusion Models"
_ICLR.cc/2025/Conference — ICLR 2025 Poster_

### Official Review · Reviewer_pwHp · 2024-11-01

**Soundness:** 3
**Presentation:** 3
**Contribution:** 3
**Rating:** 6
**Confidence:** 2

**Summary:**

The paper presents a novel approach for erasing inappropriate concepts from text-to-image diffusion models without the need for fine-tuning. The authors identify that these models, while capable of generating sophisticated images, can inadvertently produce content that is ethically and legally problematic, such as NSFW material and copyright-infringing styles. The proposed method GIE addresses this by injecting growth inhibitors based on identified features related to inappropriate concepts during the diffusion process. An adapter is also trained to infer the suppression scale, accommodating varying degrees of inappropriateness. The paper claims that GIE effectively captures subtle word manifestations at the image level, enabling precise erasure of target concepts without affecting other concepts or image quality.

**Strengths:**

- The writing is clear and easy to follow.
- The discussed topic and motivation are both innovative and significant.
- The proposed method can erase inappropriate concepts from text-to-image diffusion models without the need for fine-tuning.

**Weaknesses:**

- Some previous works on LLMs for enhancing diffusion models, such as [1][2][3][4], should be included in the related work section.
- I'm curious about the explicit or implicit damage to the model itself when inappropriate concepts are removed. Generally, inappropriate concepts may not be entirely independent of some "appropriate concepts." This analysis needs to be considered.
- What is the cost of using GPT-4o for data processing in this paper? For instance, the expenses and time involved. Such information should be listed to provide valuable insights to the community.





[1] SUR-adapter: Enhancing text-to-image pre-trained diffusion models with large language models

[2] PromptCrafter: Crafting Text-to-Image Prompt through Mixed-Initiative Dialogue with LLM

[3] LLM Blueprint: Enabling Text-to-Image Generation with Complex
and Detailed Prompts.

[4] LLM-grounded Diffusion: Enhancing Prompt Understanding of Text-to-Image Diffusion Models with Large Language Models.

**Questions:**

see weakness

---

> ### Author Response · Authors · 2024-11-21
> **Response (1/1) to Review pwHp**
>
> Thank you for the thoughtful and constructive feedback that helped us improve our work. In response to the issues you raised, we have made clarifications below.
>
> **Response to W1:** The main topics of these four papers are improving the quality and content of prompts to enrich the details of the generated images. And the purpose of our work is to prevent the text-to-image diffusion model from generating inappropriate images. We think these four works are rarely relevant to our work and we believe that the most likely scenario to apply these four works to our task is to reduce the possibility of introducing unsafe information by refining the input prompt.
>
> However, our GIE can even solve the problem that implicit unsafe prompts may lead to inappropriate images. Locating and identifying unsafe features in images from an image perspective, the growth inhibitors suppress the expression of these features, thereby avoiding the generation of unsafe images by both explicit and implicit unsafe prompts. Compared with purifying input prompts, our method further eliminates the generation of unsafe images.
>
> We have added the content mentioned here in the related work section, as shown in Lines 150-153.
>
> ---
>
> **Response to W2:** Since our GIE does not fine-tune the model, it will not affect the model's cognition and will not cause catastrophic forgetting. In the experiment, we verified the FID and CLIP scores to show that our GIE has almost no impact on the quality and semantics of the image (see Table 1).
>
> We can analyze the damage to the model from the images generated by other methods. For example, ESD will replace the concept of nakedness with the concept of bed, and the person can no longer be seen in the picture (Figure 18). SalUn, an erasure method, is very affected by the training set. Since the people in many naked pictures are standing, the training is easy to overfit, resulting in the same posture as the people in the generated pictures.
>
> + **Inappropriate concepts may not be entirely independent of some "appropriate concepts."**
>
> We have considered this issue and showed that other concepts are related to the target concept in illustrative examples (such as Figures 1 and 3, where "nudity" is inevitably related to "person"). In fact, our method can resolve it in a better way than other methods by accurately locating and removing inappropriate concepts using the attention mechanism with minimal effect on other concepts (e.g., our method preserves the semantic concept "person", whereas other methods may replace "person" with other objects for nudity erasure).
>
> ---
>
> **Response to W3:** For one concept, we only need to obtain its intermediate values and let GPT-4o label the desired suppression scales for images generated from 60 prompts (approximately $1, in 20 minutes) and then train a two-layer MLP (about 30 seconds). Therefore, the training cost of the GIE method is low. We have added th above information to Appendix D.

---

> > ### Comment · Reviewer_pwHp · 2024-11-21
> > **Thank you for your reply**
> >
> > Thank you for your reply. All my concerns have been solved and I tend to raise my score.

---

> ### Author Response · Authors · 2024-11-23
> **Thanks for the Feedback**
>
> Dear Reviewer pwHp,
>
> We are pleased to hear that you no longer have concerns. Thank you again for the time and effort you put into reviewing our paper.
>
>
>
> Best regards,
>
> Authors

---

### Official Review · Reviewer_1Bko · 2024-11-01

**Soundness:** 3
**Presentation:** 3
**Contribution:** 4
**Rating:** 6
**Confidence:** 4

**Summary:**

This paper proposes a new method GIE that injects growth inhibitors into the attention map to erase certain harmful concepts. The method can be applied without fine-tuning, with the injection level controlled by an auxiliary network. Experimental results show the effectiveness of GIE in erasing nude concepts while preserving the image quality.

**Strengths:**

- The main strength of the proposed method is that it does not require fine-tuning, which is highly beneficial in terms of efficiency. Additionally, the method is widely applicable to architectures similar to Stable Diffusion.
- Most of the proposed methods are based on observations that support the validity of the approach.
- Compared to the baselines, GIE erases the harmful and other concepts exceptionally well.

**Weaknesses:**

- While most methods are based on observations, I feel the paper lacks *sufficient intuition* or theoretical justification for each proposed method. How does inserting an attention map for concept erasure lead to the removal of specific concepts? What is the meaning of these new attention maps in M_replace? How does this reweighted attention map function as a growth inhibitor? It is good that you avoid the naive approach of simply suppressing the most related attention map, but since your solution deviates from previous attention modulation methods, while we can see that it works, it is challenging to understand why it works so effectively.
- It appears that the adapter learns a function to map intermediate values to suppression values. However, intermediate values vary significantly with different text prompts. Although the authors demonstrate the generalizability of learned prompts, it is questionable whether training the adapter on a limited dataset will generalize effectively when the test set encompasses a broader range of concepts.
- The model's applicability is currently limited to Stable Diffusion 1.4. Since the method relies on the *specific architecture*, I believe additional experiments on different Stable Diffusion models would better showcase its effectiveness.
- This also relates to weakness #3: although GIE is effective and efficient, it requires a *high level of engineering*, such as designing an adaptive model and correctly injecting features. The optimal range of weights (w), suppression levels, and injection positions might vary depending on the target concepts or diffusion models. It would have been beneficial to demonstrate that the same method, with similar configurations, performs well on other models.

**Questions:**

- In line 415 (416?), how could the CLIP model be leveraged to determine whether specific objects exist in the generated images?
- Do the NSFW removal rates used as a metric refer to the ratio averaged over test prompts calculated using NudeNet?

---

> ### Author Response · Authors · 2024-11-21
> **Response (1/3) to Review 1Bko**
>
> Thank you for the thoughtful and constructive feedback that helped us improve our work. In response to the issues you raised, we have made clarifications below.
>
> **Response to W1:** Thanks for the helpful comments. Based on these comments, we have added Appendix A to provide a detailed justification of the proposed method and Appendix B to carry out ablation studies on the injection position of growth inhibitors in the revised paper.
>
> + **Clarification on reweighted attention map and growth inhibitor:** As shown in previous studies [3,4,5], the attention map contains the image features that need to be paid attention to, and different values in the attention map signify the importance of different pixels. To locate the region of attention corresponding to the target concept $ p^* $, we obtain its attention map group $ M^* $. If we multiply the target attention map by a negative weight, we can turn the region that requires more attention into the region that needs more suppression. Therefore, the attention maps after multiplying the negative weight can be considered growth inhibitors.
> + **Rationale behind the insertion of attention map for concept erasure:** The elements in the attention map group are mapped with the elements in the text embedding one-to-one. Different from the information in the text embedding that is affected by order  (see [1,2], the transformer structure allows the subsequent token to contain the information of the previous token), the attention maps obtained interact with each other in Unet and jointly affect the generation of images. For this reason, additional injected image features (growth inhibitors) can also play a role in the generation process and guide it in the opposite direction. We conducted an experiment to further illustrate the above point.  Let the prompt be "a happy woman and a sad man" and encode it, we swap the corresponding positions of "happy" and "sad" in its text embedding and generate an image, and find that the result is exactly the same as the original image, as shown in Figure 9. This proves that the order of each item in the text embedding does not affect the generated result and jointly affects the generated results. Therefore, we can infer that the additionally introduced attention maps (growth inhibitors) will also work together with the original attention maps during image generation.
> + **The meaning of new attention maps in M_replace:** $ M_{replace} $ obtained after adding the growth inhibitor replaces the original $ M $ and enters the subsequent procedure. After clarifying the above content, we can infer that the position to inject growth inhibitors has little effect on the result. Here we inject it between [SOT] and [EOT] to retain their original role of indicating the beginning and end of the text sequence, and we choose the position preceding [EOT] for ease of implementation.
>
> ---
>
> **Response to W2:** The proposed adaptor demonstrates a certain level of generalizability. For example, after learning the inhibition level of cats, it can be extended to the erasure task of dogs. This may be attributed to the substantial similarities in recognizable image features shared between the two tasks.
>
> Specifically, the adapter effectively learns a mapping function that correlates intermediate values to suppression scales. Here, the intermediate values are derived from the image features pertinent to the target concept, while the suppression scales serve to re-weight these image features. Taking the task of erasing the concept of "nude" as an example, despite differences in the original prompts, our intermediate values all come from $ m^*_{nude} $, which contain a large number of image features of the target concept. We found that these image features are associated with the suppression scales and this association can be captured by the adapter (as illustrated in Figure 3).
>
> For target concepts exhibiting significantly different characteristics, the generalizability may be limited. In this case, we recommend adding this new target concept and retraining the MLP. In the future, it may be beneficial to explore using a more sophisticated model to address this issue. However, for our current erasure tasks, such as NSFW and style, the two-layer MLP is sufficient in most cases.

---

> > ### Author Response · Authors · 2024-11-21
> > **Response (2/3) to Review 1Bko**
> >
> > **Response to W3:** Thanks for the comments on the experiments. We have added Appendix G in the revised paper to show the effectiveness of GIE on different versions of Stable Diffusion. GIE in different stable diffusion models can effectively erase naked concepts and obtain excellent NRR results without affecting semantics and quality, and also perform well in FID and CLIP scores. Since different stable diffusion models have different intermediate values, we train the corresponding versions of the adapter and test our method on the I2P dataset and the COCO-30K dataset as following tables (see Figure 16 and Table 5 in Appendix G). We also include more visual examples in Figure 17.
> >
> >
> > | **Number of NSFW images on I2P dataset** |         |                       |                |         |                       |                |       |                     |               |
> > |--------------------------------------|---------|-----------------------|----------------|---------|-----------------------|----------------|-------|---------------------|---------------|
> > | **Exposed Body Parts**                   | **SD v1.5** | **GIE (\*Ours)on SD v1.5** | **NRR on SD v1.5** | **SD v2.1** | **GIE (\*Ours)on SD v2.1** | **NRR on SD v2.1** | **SD XL** | **GIE (\*Ours)on SD XL** | **NRR on SD XL**  |
> > | Total                                | 1059    | 184                   | -82.63%        | 574     | 195                   | -66.03%        | 824   | 355                 | -56.92%       |
> > | Buttocks                             | 67      | 10                    | -85.07%        | 28      | 13                    | -53.57%        | 28    | 7                   | -75.00%       |
> > | Female_Breast                        | 235     | 20                    | -91.49%        | 135     | 30                    | -77.78%        | 128   | 39                  | -69.53%       |
> > | Female_Genitalia (40)                | 31      | 3                     | -90.32%        | 9       | 3                     | -66.67%        | 22    | 6                   | -72.73%       |
> > | Armpits                              | 295     | 52                    | -82.37%        | 148     | 53                    | -64.19%        | 297   | 140                 | -52.86%       |
> > | Belly                                | 230     | 46                    | -80.00%        | 153     | 54                    | -64.71%        | 165   | 88                  | -46.67%       |
> > | Male_Breast                          | 57      | 8                     | -85.96%        | 26      | 13                    | -50.00%        | 37    | 20                  | -45.95%       |
> > | Male_Genitalia                       | 29      | 11                    | -62.07%        | 15      | 6                     | -60.00%        | 12    | 6                   | -50.00%       |
> > | Feet                                 | 115     | 34                    | -70.43%        | 60      | 23                    | -61.67%        | 135   | 49                  | -63.70%       |
> >
> >
> >
> > | **Metric**         | **GIE (\*Ours)on SD v1.5** | **GIE (\*Ours)on SD v2.1** | **GIE (\*Ours)on SD XL**  |
> > |----------------|----------------|----------------|---------------|
> > | FID (↓)        | 15.995         | 16.029         | 16.481        |
> > | CLIP score (↑) | 25.121         | 24.214         | 26.431        |
> >
> > ---
> >
> > **Response to W4:** The training process of the proposed GIE is relatively straightforward, as the adaptor adopted is a two-layer MLP. The main training effort in our GIE is to establish the connection between the target concept's intermediate values and its suppression scales (let GPT-4o label the desired suppression scales). And the injection position is found to be irrelevant to the erasure result (as shown in Appendix B) and does not affect the training process. For one concept, we only need to obtain its intermediate values and let GPT-4o label the desired suppression scales for images generated by 60 prompts (which incurs a cost of approximately $1 and about 20 minutes), and subsequently train a two-layer MLP (about 30 seconds). Therefore, our entire training process is lightweight. In our response to W3, we have demonstrated the effectiveness of GIE across different Stable Diffusion models.

---

> > > ### Author Response · Authors · 2024-11-21
> > > **Response (3/3) to Review 1Bko**
> > >
> > > **Response to Q1:** We provide details on classification with CLIP in Appendix H. The CLIP model can be used for zero-shot image classification tasks because it maps the embedding vectors of images and texts into the same space so that the similarity between images and texts can be measured by their cosine similarity. Therefore, for our task of deciding whether a specific object exists in an image, we use the text template "a photo of a [class]" and substitute 10 classes in CIFAR-10 into [class] to get 10 prompts. We encode these 10 prompts and images for the classification with CLIP and calculate the cosine similarity between the image embedding and each text embedding. The text embedding with the highest cosine similarity to the image is classified as its object category.
> > >
> > > ---
> > >
> > > **Response to Q2:** NSFW removal rate (NRR) **does not** refer to the average rate of test prompts calculated using NudeNet. If the number of images containing exposed body parts generated after applying the erasure method is $ n_{\text{erase}} $, and the number generated by the original model, such as SD v1.4, is $ n_{\text{original}} $, NRR is the ratio of $ n_{\text{original}} $ to $ n_{\text{erase}} $ by$ NRR = \frac{n_{original}-n_{erase}}{n_{original}} $. We added the details in Lines 417-420 of the revised paper.
> > >
> > > ---
> > >
> > > **References**
> > >
> > > [1] Radford, Alec, et al. "Learning transferable visual models from natural language supervision." International conference on machine learning. PMLR, 2021.
> > >
> > > [2] Raffel, Colin, et al. "Exploring the limits of transfer learning with a unified text-to-text transformer." Journal of machine learning research 21.140 (2020): 1-67.
> > >
> > > [3] Hong, Susung, et al. "Improving sample quality of diffusion models using self-attention guidance." Proceedings of the IEEE/CVF International Conference on Computer Vision. 2023.
> > >
> > > [4] Hertz, Amir, et al. "Prompt-to-prompt image editing with cross attention control." The International Conference on Learning Representations. 2023.
> > >
> > > [5] Liu, Bingyan, et al. "Towards Understanding Cross and Self-Attention in Stable Diffusion for Text-Guided Image Editing." Proceedings of the IEEE/CVF Conference on Computer Vision and Pattern Recognition. 2024.

---

> ### Comment · Reviewer_1Bko · 2024-11-25
> **Response to Rebuttal**
>
> Thank you for your constructive response.
>
> A few remaining questions:
>
> - If text order does not matter, could this issue stem from limitations in the text embedding's ability to consider order? Would higher-quality models with better text encoders yield different results?
> - I didn’t quite understand how the fact that order does not matter relates to the insertion of the attention map to erase a concept.
> - Perhaps I missed your point, but regarding your explanation of CLIP usage, isn’t it possible that the target object exists but doesn’t appear as the most similar to the image if there is another object in the image? Also, if there are multiple objects or if the concept is erased and no object exists, wouldn’t the result be a uniform distribution? In such cases, the target object might or might not be the most similar (Random max similarity). I wonder whether this is a widely adopted method.

---

> ### Author Response · Authors · 2024-11-25
> **Response to New Questions**
>
> We are pleased to receive your feedback and questions, and thank you for your efforts and time. We have made clarifications below.
>
> **Response to Q1:** The order of **words** in the prompt is important for the generated image, while the order of **tokens** in the text embedding is irrelevant for the generated image. So higher-quality models with better text encoders come to the same conclusion.
> Let's use an example for illustration. If we encode $ p_1 $= "a happy woman and a sad man" to $ c_1 $ and encode $ p_2 $ = "a sad woman and a happy man" to $c_2$, $ c_{happy} $ in $ c_1 $ and $ c_2 $ is different because the order of words affects the text encoder's results. And the $ image_1 $ obtained by $ p_1 $ and the $ image_2 $ obtained by $ p_2 $ are different. If we encode $ p_1 $ = "a happy woman and a sad man" to $ c_1 $ , and we swap the positions of $ c_{happy} $ and $ c_{sad} $ in $ c_1 $ to obtain $ c_3 $. Since the content in $ c_{happy} $ does not change but only its position is changed, the image results of the $ c_1 $ and $ c_3 $ are consistent.
>
> ---
>
> **Response to Q2:** The fact that the order does not matter has nothing to do with the effect of erasing concepts, but only serves to illustrate that we can insert the growth inhibitor at any position. The reason for the concept erasure effect is that we obtain the growth inhibitor and inject it into the attention map group of the original prompt. The attention maps in the attention map group reflect the image features that need to be paid attention to, and collectively shape the resulting image. Therefore, the growth inhibitors (which can be seen as attention maps) we inject enable unsafe features in the image to be fully discovered and suppressed by weighted negative values and affect the generated image together with the attention map group of the original prompt.
>
> ---
>
> **Response to Q3:** Using the CLIP model to detect whether the target object is in the image is indeed a widely adopted method [1,2].  This is because CLIP is a zero-shot pre-trained model with high accuracy.
>
> + **If there is another object in the image, is it possible that the target object exists but does not look most similar to the image?**
>
> Yes, it is possible. This is related to the accuracy of the classification model. We cannot guarantee that the classification model can achieve 100% recognition accuracy.
>
> + **If there are multiple objects or if the concept is erased and no object exists, wouldn't the result be a uniform distribution?**
>
> _Multiple objects exists:_ Since the prompt  template for generating images only contains one class, there will not be multiple categories of objects in the image, and multiple objects of the same category can also be recognized by our method.
>
> _No objects exists_:  Sorry that our explanation is not clear enough and we add more details about object recognition. After computing cosine similarity between the the image with no object and 10 prompts ("a photo of a [class]"), we obtain 10 lower results (such as [24.271, 24.272, 24.273, 24.274, 24.275, 24.276, 24.277, 24.278, 24.279, 24.274]), which will enter the softmax to obtain 10 specific category probabilities ([0.0996, 0.0997, 0.0998, 0.0999, 0.1000, 0.1001, 0.1002, 0.1003, 0.1004, 0.0999]). If the highest category probability (0.1004) is lower than the threshold, we imply that objects in the image have been erased.
>
> ---
>
> **Reference:**
>
> [1] Lyu, Mengyao, et al. "One-dimensional Adapter to Rule Them All: Concepts Diffusion Models and Erasing Applications." Proceedings of the IEEE/CVF Conference on Computer Vision and Pattern Recognition. 2024.
>
> [2] Kumari, Nupur, et al. "Ablating concepts in text-to-image diffusion models." Proceedings of the IEEE/CVF International Conference on Computer Vision. 2023.

---

### Official Review · Reviewer_yvQD · 2024-11-03

**Soundness:** 2
**Presentation:** 2
**Contribution:** 2
**Rating:** 6
**Confidence:** 3

**Summary:**

Authors focuses on a concept erasing problem which is caused by explicit NSFW prompts and importantly implicit unsafe prompts. Authors propose to inject "Growth Inhibitors", which is a slice of attention map group of target concept, to current attention map group. The re-weighting scheme of "Growth Inhibitors" are provided by proposed Suppression Scale Adapter.

**Strengths:**

1) The proposed method doesn't require any fine-tuning concentrated on cross-attention layers -- which is the general methodology in the recent works in concept erasing literature, therefore, this work distinguishes itself from the other works.
2) By training a Suppression Scale Adapter, the proposed method does not require additional training/finetuning for each different concepts.
3) As shown in some qualitative experiments, after concept erasing operation, most of the spatial details are preserved -- especially when compared to other methods.

**Weaknesses:**

1) There is no clear declaration of why injected Growth Inhibitors I is inserted right before [EOT]. There is no ablation study on the position of Growth Inhibitors. More clarifications and motivations behind this, and ablations will help us to understand the proposed method better.

2) No quantitative experimentation on multi-concept erasing. Furthermore, no qualitative results for more than 2 concept erasing. This makes it difficult to assess the real performance of this method.

3) Qualitative results shows that, proposed method can only erase semantic concept up to a certain extend. For instance, in Figure 7 Bottom row, when erasing Van Gogh and Nude concepts, the resultant image still preserves the style of Van Gogh depicted by the color schema. This problem is also seen in Figure 13.

**Questions:**

1) Proposed method contradicts with the statement in Line 252: "A naive approach to erasure is to weaken tokens whose attention maps are similar to the extracted features. Unfortunately, this method might not work because...". The reason is the following:
- First, original Attention Map Groups are injected with the Growth Inhibitor I.
- Second, the attended representation MV is calculated based on injected Attention Map Group M'.
- As a result, Final output of Cross-Attention is calculated with M'V.
- In SD v1.4, Value (V) is a linear projection of token embeddings c.
- Considering all of these mentioned properties, the final output reduces to calculating Final Output = (M_[SOT] x V) + ... + (M*_[target_concept] x V) + (M_[EOT] x V)
- As this final output calculation demonstrates, (M*_[target_concept] x V) directly corresponds to weakening every single token embedding -- since V is a function of token embedding c.

It would be good to clarify this.

2) What happens when the concept prompt has more than 1 tokens? For example, when erasing the concept of Van Gogh, the token embeddings has the form <c_[SOT], c_[Van], c_[Gogh], c_[EOT] >. How to reduce c_[Van], c_[Gogh] into one, unified representation? Is Simultaneous Suppression of Multiple Concepts strategy described in Section 4.3 applied here?

---

> ### Author Response · Authors · 2024-11-21
> **Response (1/2) to Review yvQD**
>
> Thank you for the thoughtful and constructive feedback that helped us improve our work. In response to the issues you raised, we have made clarifications below.
>
> **Response to W1:** Thanks for the helpful comment. In the revised paper, we have added Appendix A to provide a detailed clarification of the injection position of growth inhibitors and carried out ablation studies in Appendix B.
>
> + **Clarification on the position of Growth Inhibitors (Appendix A).**
>
> In general, due to the attention mechanism in the Unet of the diffusion model, the additional attention maps (growth inhibitors) can work together with the original attention maps during image generation, and thus the location where Growth Inhibitors are injected has no obvious effect on the erasure results. We can inject it anywhere between [SOT] and [EOT] so that [SOT] and [EOT] still indicate the beginning and end positions of the text sequence. And we chose the position preceding [EOT] for ease of implementation.
>
> + **Ablation study on the position of Growth Inhibitors (Appendix B).**
>
> The above clarification is further confirmed by the ablation study in Appendix B. For the task of erasing the concept of "nude", we compare the results of GIE when injecting growth inhibitors in different positions (before $ m_\text{[EOT]} $, after $ m_{\text{[SOT]}} $, and at a random position). We show the NRR results on the I2P dataset as follows (see also Figure 10 in Appendix B) and find that these three variants yield the same erasing effects.
>
>
> |  **NRR Results on I2P Dataset** ||||
> | --- | --- | --- | --- |
> | **Exposed Body Parts (in SD v1.4)** | **GIE (\*Ours,before $m_{[EOT]}$)** | **after $ m_{[SOT]} $** |**random position** |
> | Total (1037) | -77.05% | -77.05% | -77.05%|
> | Buttocks (70) | -87.14% | -87.14% |-87.14%|
> | Female_Breast (236) | -87.71% | -87.71% |-87.71%|
> | Female_Genitalia (32) | -90.63% | -90.63% |-90.63%|
> | Armpits (287) | -72.82% | -72.82% |-72.82%|
> | Belly (215) | -67.91% | -67.91% |-67.91%|
> | Male_Breast (56) | -82.14% | -82.14% |-82.14%|
> | Male_Genitalia (36) | -69.44% | -69.44% |-69.44%|
> | Feet (105) | -72.38% | -72.38% |-72.38%|
>
>
> ---
>
> **Response to W2:** Thanks for the comment on our experiments. We have conducted additional quantitative and qualitative experiments for multi-concept erasing in Appendix F of the revised paper.
>
> + **Quantitative experimentation on multi-concept erasing**
>
> To erase the two concepts "nude" and "Van Gogh" simultaneously, we generate 200 images using "a painting of a nude person in Van Gogh Style" and use NudeNet to detect the exposed parts. As shown in the following table (see also Figure 14 in Appendix F), we have almost completely erased the concept of "nude". Subsequently, we use GPT-4o to classify the styles of these two hundred images and find that only three images are classified as Van Gogh style. This shows the effectiveness of our approach in maintaining the superior erasing effect for multiple concepts.
>
> | **NRR Results**                     |              |
> |---------------------------------|--------------|
> | **Exposed Body Parts (in SD v1.4)** | **GIE (\*Ours**  |
> | Total (398)                     | -99.50%      |
> | Buttocks (26)                   | -100.00%     |
> | Female_Breast (141)             | -99.29%      |
> | Female_Genitalia (40)           | -100.00%     |
> | Armpits (81)                    | -100.00%     |
> | Belly (103)                     | -99.03%      |
> | Male_Breast (1)                 | -100.00%     |
> | Male_Genitalia (3)              | -100.00%     |
>
>
> + **Qualitative results for more than 2 concept erasing.**
>
> We also show the generated examples when erasing more than two concepts in Figure 15 of the revised paper. We discussed multi-concept erasing across three scenarios and showed that GIE achieved excellent erasure results in all these scenarios:
> (1) The image contains all the target concepts.
> Example 1: prompt = "This is a _Van Gogh_ style painting of a _dog_ and a _cat_ playing with a _naked_ woman on the beach."; target concepts= "_Van Gogh_", "_nude_", "_dog_", "_cat_".
> (2) The image contains some of the target concepts.
> Example 2: prompt = "This is a _Van Gogh_ style painting of a _naked_ woman on the beach."; target concepts="_Van Gogh_", "_nude_", "dog", "cat".
> Example 3: prompt = "This is a _Van Gogh_ style painting of a _naked_ woman on the beach."; target concepts= "_nude_", "dog", "cat", "bike".
> Example 4: prompt = "This is a _Van Gogh_ style painting of a _naked_ woman on the beach.v; target concepts=  "_Van Gogh_", "dog", "cat".
> (3) The image does not contain any target concepts.
> Example 5: prompt = "This is a _Van Gogh_ style painting of a _naked_ woman on the beach."; target concepts="dog", "cat", "bike".

---

> > ### Author Response · Authors · 2024-11-21
> > **Response (2/2) to Review yvQD**
> >
> > **Response to W3:** We think that in the images you pointed out, Van Gogh's brushstrokes and features have been mostly erased, and the preservation of the color schema alone does not imply that the style erasure was insufficient.
> >
> > Our method is designed to preserve essential details of the image to maintain its semantic integrity; consequently, the color may remain relatively unchanged during the style-erasing task. This may stem from the generative model's assessment that color is not a defining characteristic of the style. However, if there is a specific style exhibiting recognizable color features, it can also be effectively erased through GIE. As illustrated in Figure 7 and Figure 19 (original Figure 13), the brushstrokes in the image after GIE erasure became delicate and smooth, and the swirling curves characteristic of van Gogh were lost, making the image closer to the style of flat painting and realism.
> >
> > Moreover, if the current level of suppression is deemed insufficient, one can address this issue by retraining the adapter. By setting detailed system prompts for GPT-4o, the suppression scale annotated by GPT-4o can be enhanced, resulting in a greater difference between the generated images and the original images. Here, we refine the description of Van Gogh's style, let GPT-4o re-annotate the suppression scale, use it to train a new style adapter, and apply it in Figure 15.
> >
> > ---
> >
> > **Response to Q1:** Regarding the reasoning process you presented, you seem to illustrate that the injection of growth inhibitors in GIE is also a way to "weaken tokens". The original phrasing in our paper may have led to potential misunderstandings. To enhance clarity and avoid misunderstandings, we changed Lines 253-254 to "A naive approach to erasure is to weaken tokens in the prompt $ p $ whose attention maps are similar to the extracted features from $ M^* $".
> >
> > The naive approach mentioned here is to weaken the tokens in the original prompt, which will affect other concepts (as shown on the left side of Figure 3), while our proposed GIE specifically targets and weakens the additional growth inhibitor introduced, which accurately locates the target concept and erases it.
> >
> > ---
> >
> > **Response to Q2:** If $ c^* = [c^*_{\text{[SOT]}},c^*_{\text{Van}},c^*_{\text{Gogh}},c^*_{\text{[EOT]}}] $, then we extract $ m^*_{Van} $ and $ m^*_{Gogh
> > } $ and assign them negative weights, i.e., $ w_1 $ and $ w_2 $, respectively, to obtain $ w_1·m^*_{Van} $ and $ w_2·m^*_{Gogh} $. There is no multi-concept strategy involved here because these two tokens appear in the same concept. In actual applications, the two tokens "Van" and "Gogh" respectively learn the common elements and brushstrokes of this style, and have different semantic information, so the suppression scales will be set separately.

---

> ### Author Response · Authors · 2024-11-23
> **Discussion Inquiry**
>
> Dear Reviewer yvQD,
>
> Many thanks for your efforts in reviewing this paper. Your careful reading and insightful comments indeed help us a lot in improving this paper. We have provided responses to your questions and the weaknesses you mentioned. We hope this can address your concerns.
>
> If our responses satisfactorily address your concerns, we would like to kindly ask if you could consider raising the score. If not, please let us know any remaining questions you may have. We look forward to hearing from you soon.
>
> Best regards,
>
> Authors

---

> > ### Author Response · Authors · 2024-11-25
> >
> > Dear Reviewer yvQD,
> >
> > We are encouraged to see that Reviewers S8KB and pwHp have indicated their concerns have been addressed and have raised their final ratings. With only one day remaining in the discussion period, we would greatly appreciate it if you could take a moment to read our responses. We want to know whether your concerns have been addressed or if you have any further comments. Thank you for your time and effort regarding this paper.
> >
> > Best regards,
> >
> > Authors

---

> ### Author Response · Authors · 2024-11-30
>
> Dear Reviewer yvQD,
>
> Sorry to bother you again. We would like to know whether the responses have addressed your concerns.
>
> If you have any remaining questions or require further clarification, we would be more than happy to provide additional details.
> If our responses satisfactorily address your concerns, we would like to kindly ask if you could consider raising the score.
>
> Your support would mean a great deal to us and would greatly encourage our continued efforts in this area.
> Thank you once again for your time, effort, and constructive comments!
>
> Best regards,
>
> Authors

---

> > ### Comment · Reviewer_yvQD · 2024-12-01
> >
> > Thank you for addressing my concerns in the rebuttal, I am increasing my score to 6.

---

> > > ### Author Response · Authors · 2024-12-02
> > > **Thanks for the Feedback**
> > >
> > > Dear Reviewer yvQD,
> > >
> > > We are grateful for your feedback and are pleased to hear that you no longer have concerns. Thank you again for the time and effort you put into reviewing our paper.
> > >
> > > Best regards,
> > >
> > > Authors

---

### Official Review · Reviewer_S8KB · 2024-11-05

**Soundness:** 3
**Presentation:** 2
**Contribution:** 3
**Rating:** 6
**Confidence:** 4

**Summary:**

This paper proposes GIE, a method for eliminating NSFW content in diffusion models. The authors leverage the attention map of target concepts, reweighting them to synthesize "growth inhibitors" that represent undesired content. These reweighted attention maps are then injected into the diffusion process. Additionally, an adaptor is trained to determine the suppression scale dynamically. GIE outperforms eight baseline methods and effectively removes implicit concepts like "nude."

**Strengths:**

1. This work addresses a significant issue by reducing the generation of NSFW content.
2. Extensive visualizations illustrate the effectiveness of GIE.

**Weaknesses:**

1. The writing is somewhat difficult to follow in certain sections. See questions.

2. There is a lack of ablation studies on each component of GIE, such as the role of the adaptor.

3. The paper does not discuss or compare with related methods like the one proposed in [1], which also targets implicit concept removal in diffusion models.

[1] Implicit Concept Removal of Diffusion Models, ECCV 2024.

I will consider increasing the score if my concerns are addressed.

**Questions:**

1. Could the authors clarify the purpose of the SOT and EOT tokens?
2. In Section 4, line 260, how are the starting and ending positions determined?

---

> ### Author Response · Authors · 2024-11-21
> **Response (1/2) to Review S8KB**
>
> Thank you for the thoughtful and constructive feedback that helped us improve our work. In response to the issues you raised, we have made clarifications below.
>
> **Response to W1Q1:** In our method, [SOT] and [EOT] are used as structure markers to indicate the beginning and end of a text sequence, respectively. We have explained this in Lines 249-250 of the revised paper.
>
> **Response to W1Q2:** In our method, the starting and ending positions for feature extraction are determined by the positions of $ m^*_{[SOT]} $ and $ m^*_{[EOT]} $. Specifically, let the position of $ m^∗_{[SOT]} $ in $ M^* $ be $ s $ and the position of $ m^∗_{[EOT]} $ be $ e $, the starting and end positions for extraction are $ s + 1 $ and $ e − 1 $, respectively. We clarify this in Lines 264-265 of the revised paper.
>
> ---
>
> **Response to W2:** Thanks for the comment on our experiments. We have performed additional ablation studies on the adapter and the insertion position of the growth inhibitors in Appendices C and B, respectively.
>
> + The ablation study on the adapter (see Appendix C for details) has shown that the adapter can help better balance the erasure effect and semantic preservation. Specifically, we have compared the effects of erasing the concept "nude" with a fixed suppression scale and when erasing it using our adapter. The tables below (see also Figure 11 and Table 3 in Appendix C) present the NRR and CLIP scores on the I2P dataset. Although the NRR results are better when the suppression scale w = 10, this comes at the expense of significant semantic distortion, as indicated by the CLIP scores. Meanwhile, the CLIP score with the suppression scale w = 5 is very close to GIE with the adapter, but the erasing effect is inferior. Moreover, the visual examples provided in Figure 12 also show results with different fixed suppression scales.
>
>
> |  **NRR Results on I2P Dataset** ||||
> | --- | --- | --- | --- |
> | **Exposed Body Parts (in SD v1.4)** | **GIE (\*Ours,Adapter)** | **supression scale w = 5** |**supression scale w = 10** |
> | Total (1037) | -77.05% | -62.49% | -86.31%|
> | Buttocks (70) | -87.14% | -88.57% |-94.29%|
> | Female_Breast (236) | -87.71% | -80.93% |-91.95%|
> | Female_Genitalia (32) | -90.63% | -84.38% |-90.63%|
> | Armpits (287) | -72.82% | -54.70% |-85.71%|
> | Belly (215) | -67.91% | -53.49% |-84.19%|
> | Male_Breast (56) | -82.14% | -60.71% |-89.29%|
> | Male_Genitalia (36) | -69.44% | -27.78% |-66.67%|
> | Feet (105) | -72.38% | -49.52% |-78.10%|
>
>
>
> |  **CLIP score on I2P Dataset** ||||
> | --- | --- | --- | --- |
> | **Metric** | **GIE (\*Ours,Adapter)** | **supression scale w = 5** |**supression scale w = 10** |
> | CLIP score (↑) | 27.052 | 27.124 | 26.231|
>
>
> + The ablation study on the insertion position (see Appendix B for details) reveals that the three positions (before $ m_{[EOT]} $, after $ m_{[SOT]} $, and at a random position) exhibit comparable erasing effects. Specifically, we investigate the task of erasing the concept of "nude" by comparing the performance of the Growth Inhibitor for Erasure (GIE) when injected at these three different positions. The NRR results on the I2P dataset are presented in the following table (see also Figure 10 in Appendix B). We attribute this similarity in erasing effectiveness to the collective influence of all attention maps within the Unet on the generated results, which remain invariant to the order of insertion. Further explanations can be found in Appendix B.
>
>
> |  **NRR Results on I2P Dataset** ||||
> | --- | --- | --- | --- |
> | **Exposed Body Parts (in SD v1.4)** | **GIE (\*Ours,before $m_{[EOT]}$)** | **after $ m_{[SOT]} $** |**random position** |
> | Total (1037) | -77.05% | -77.05% | -77.05%|
> | Buttocks (70) | -87.14% | -87.14% |-87.14%|
> | Female_Breast (236) | -87.71% | -87.71% |-87.71%|
> | Female_Genitalia (32) | -90.63% | -90.63% |-90.63%|
> | Armpits (287) | -72.82% | -72.82% |-72.82%|
> | Belly (215) | -67.91% | -67.91% |-67.91%|
> | Male_Breast (56) | -82.14% | -82.14% |-82.14%|
> | Male_Genitalia (36) | -69.44% | -69.44% |-69.44%|
> | Feet (105) | -72.38% | -72.38% |-72.38%|

---

> > ### Author Response · Authors · 2024-11-21
> > **Response (2/2) to Review S8KB**
> >
> > **Response to W3:** Thanks for sharing a recent work relevant to ours. This work mainly focuses more on watermarked and QR data and less on NSFW content. We have implemented their method (Geom-Erasing) and reproduced their results ourselves, as their implementation has not been publicly available yet. The results we obtained, including the NRR results on the I2P dataset and the FID and CLIP score results on the COCO-30K dataset, are shown in the following tables.
> >
> > In summary, our proposed GIE outperforms Geom-Erasing in both erasing effect and image quality. Specifically, Geom-Erasing needs to point out the specific unsafe regions in NSFW images to construct the training dataset, which requires additional effort to label the dataset, and the suppression by bin in the pixel space is relatively rough, which can cause overfitting and catastrophic forgetting during training. In contrast, our GIE accurately locates the region to be suppressed through the attention mechanism to customize and inject growth inhibitors. Guiding the target concept in the opposite direction during inference, we avoid the defects of Geom-Erasing and obtain better results in practice.
> >
> > | **Exposed Body Parts (in SD v1.4)** | **NRR Results on I2P Dataset** | |
> > | --- | --- | --- |
> > | | **GIE(\*Ours)** | **Geom-Erasing** |
> > | Total (1037) | -77.05% | -47.54% |
> > | Buttocks (70) | -87.14% | -31.43% |
> > | Female_Breast (236) | -87.71% | -16.10% |
> > | Female_Genitalia (32) | -90.63% | -12.50% |
> > | Armpits (287) | -72.82% | -72.13% |
> > | Belly (215) | -67.91% | -39.53% |
> > | Male_Breast (56) | -82.14% | -73.21% |
> > | Male_Genitalia (36) | -69.44% | -86.11% |
> > | Feet (105) | -72.38% | -49.52% |
> >
> >
> > | **Meric** | **GIE(\*Ours)** | **Geom-Erasing** |
> > | --- | --- | --- |
> > | FID ↓ | 15.452 | 17.747|
> > | CLIP score↑ | 26.433| 25.262|

---

> ### Author Response · Authors · 2024-11-23
> **Discussion Inquiry**
>
> Dear Reviewer S8KB,
>
> Many thanks for your efforts in reviewing this paper. Your careful reading and insightful comments indeed help us a lot in improving this paper. We have provided responses to your questions and the weaknesses you mentioned. We hope this can address your concerns.
>
> If our responses satisfactorily address your concerns, we would like to kindly ask if you could consider raising the score. If not, please let us know any remaining questions you may have. We look forward to hearing from you soon.
>
> Best regards,
>
> Authors

---

> > ### Comment · Reviewer_S8KB · 2024-11-25
> > **response to the rebuttal**
> >
> > Thank you for your rebuttal and my concerns are all addressed.
> > I raise my score to 6.

---

> > > ### Author Response · Authors · 2024-11-25
> > > **Thanks for the Feedback**
> > >
> > > Dear Reviewer S8KB,
> > >
> > > We are grateful for your feedback and are pleased to hear that you no longer have concerns. Thank you again for the time and effort you put into reviewing our paper.
> > >
> > > Best regards,
> > >
> > > Authors

---

### Author Response · Authors · 2024-11-21
**Official Comment by Authors**

Dear Reviewers,

We would like to express our sincere gratitude for your valuable feedback on our paper. We have provided detailed responses to each of your comments and have uploaded a revised version of the paper, with the changes highlighted in blue.

---

### Meta-Review · Area_Chair_jYy9 · 2024-12-18

**Metareview:**

**Summary of the Paper**
This paper introduces GIE, a method for erasing harmful or inappropriate concepts, including implicit NSFW content, from text-to-image diffusion models. Rather than requiring additional fine-tuning, GIE directly modifies the model’s diffusion process by injecting “growth inhibitors,” subsets of reweighted attention maps associated with unwanted concepts. These adjusted attention signals guide the diffusion model in suppressing specific content while preserving overall image quality. Experimental results demonstrate that GIE eliminates problematic visual elements, such as nude representations, without degrading other aspects of the generated images.

**Strengths:**
* The proposed method achieves effective concept removal, including implicit NSFW content, without the need for fine-tuning.
* The approach leverages a suppression scale adapter, enabling flexible, dynamic control over erasure intensity across various architectures.
* The authors provide extensive qualitative results that show preserved spatial details and superior concept removal compared to baseline methods.

**Weaknesses:**
* The adapter’s generalizability to target concepts that differ significantly from those seen during training remains uncertain, potentially requiring retraining and limiting its applicability across a broader concept space.
* The paper lacks sufficient descriptions and motivations for specific components of the proposed method, such as structure markers and attention maps. However, the authors provided comprehensive explanations during the rebuttal period, addressing these concerns.
* The experiments did not include essential evaluations, such as ablation studies, assessments across various model versions, and multi-concept erasing experiments. In response, the authors conducted and presented additional experiments during the rebuttal phase.

**Main Reasons for the Decision**
The paper tackles a critical problem, erasing NSFW content from text-to-image diffusion models and offering a novel approach that differentiates it from prior methods. Reviewers initially raised concerns about the lack of an ablation study, evaluations of different models, and more precise explanations of the proposed method. However, the authors provided comprehensive clarifications during the rebuttal phase and conducted additional experiments. These efforts have alleviated the primary reservations, leaving the reviewers in agreement that the paper now offers both conceptual advancements. Given the strengthened evidence, improved clarity, and responsiveness to feedback, the consensus is that the paper merits acceptance.

**Additional Comments On Reviewer Discussion:**

During the rebuttal period,
* “S8KB” and “yvQD” mentioned the need for ablation studies on the components of GIE. In response, the authors conducted ablation studies and included the results in Appendix B and C.
* “S8KB,” “yvQD,” and “1Bko” requested more precise explanations and justifications for several methodological components, including structure markers and attention maps. The authors addressed this by providing a more detailed rationale for the proposed approach, adding further clarifications in the Method and Appendix sections.
* “1Bko” emphasized the importance of evaluating the method on models beyond Stable Diffusion 1.4. The authors conducted further experiments on Stable Diffusion v1.5, v2.1, and XL, demonstrating the method’s broader applicability. These new results are presented in Appendix G.
* “S8KB” and “pwHp” suggested considering additional related works. The authors incorporated these suggested references into their related works section and implemented a method (Geom-Erasing) for further comparison, showing that GIE outperforms.

Considering these revisions and additional experiments, most reviewers indicated that their concerns had been addressed, leading to increased scores from “S8KB,” “yvQD,” and “pwHp.”

---

### Decision · Program_Chairs · 2025-01-22

Accept (Poster)